# Potentiation of P2RX7 as a host-directed strategy for control of mycobacterial infection

**Molly A Matty[1,2], Daphne R Knudsen[1], Eric M Walton[1], Rebecca W Beerman[1], Mark R Cronan[1], Charlie J Pyle[1], Rafael E Hernandez[3,4], David M Tobin[1,5]\***

[1]Department of Molecular Genetics and Microbiology, Duke University School of Medicine, Durham, United States; [2]University Program in Genetics and Genomics, Duke University, Durham, United States; [3]Center for Global Infectious Disease Research, Seattle Children's Research Institute, Seattle, United States; [4]Department of Pediatrics, University of Washington, Seattle, United States; [5]Department of Immunology, Duke University School of Medicine, Durham, United States

**Abstract** *Mycobacterium tuberculosis* is the leading worldwide cause of death due to a single infectious agent. Existing anti-tuberculous therapies require long treatments and are complicated by multi-drug-resistant strains. Host-directed therapies have been proposed as an orthogonal approach, but few have moved into clinical trials. Here, we use the zebrafish-*Mycobacterium marinum* infection model as a whole-animal screening platform to identify FDA-approved, host-directed compounds. We identify multiple compounds that modulate host immunity to limit mycobacterial disease, including the inexpensive, safe, and widely used drug clemastine. We find that clemastine alters macrophage calcium transients through potentiation of the purinergic receptor P2RX7. Host-directed drug activity in zebrafish larvae depends on both P2RX7 and inflammasome signaling. Thus, targeted activation of a P2RX7 axis provides a novel strategy for enhanced control of mycobacterial infections. Using a novel explant model, we find that clemastine is also effective within the complex granulomas that are the hallmark of mycobacterial infection.
DOI: https://doi.org/10.7554/eLife.39123.001

**\*For correspondence:**
david.tobin@duke.edu

**Competing interests:** The authors declare that no competing interests exist.

## Introduction

Tuberculosis killed approximately 1.6 million people in 2017, with around 10 million new active cases (*World Health Organization, 2018*). The continued difficulty in treatment arises, in large part, from long and relatively complex treatment regimens, currently requiring at least 6 months of treatment with multiple antibiotics (*World Health Organization, 2018*). Failures in treatment and poor patient adherence have driven the emergence of drug-resistant strains that present further therapeutic and public health challenges. Host-directed therapies (HDTs) are a new conceptual approach designed to enhance host-mediated clearance of bacteria or limit immunopathology rather than targeting bacterial processes directly (*Clatworthy et al., 2018*; *Hawn et al., 2013*). By targeting host processes, HDTs present an orthogonal approach to existing antibiotics. Notably, they avoid the danger of commonly arising bacterial mutations that eliminate binding of the drug to a bacterial target. Although a number of potential HDTs for tuberculosis (TB) have been identified in cell-culture-based models and directed animal studies, few have advanced to clinical trials (*Clatworthy et al., 2018*; *Hawn et al., 2013*).

The complex lifestyle of *Mtb* renders it particularly amenable to host-directed therapies at multiple points in its lifecycle, providing a number of druggable host targets (*Stanley et al., 2014*; *Sundaramurthy et al., 2013*). Once inside host macrophages, *Mtb* evades and exploits

macrophage-specific defense mechanisms, induces formation of characteristic aggregates called granulomas (*Pagán and Ramakrishnan, 2018*), and can persist in the face of an active immune response (*Philips and Ernst, 2012*). Virulent mycobacteria actively evade a number of cell-autonomous defense pathways (*Philips and Ernst, 2012*); mycobacteria have been reported to inhibit phagolysosome fusion (*Tan and Russell, 2015*), manipulate cell death pathways for their own survival (*Srinivasan et al., 2014*), mediate resistance to oxidative stress (*Nambi et al., 2015*), as well as survive at low pH in fully acidified phagolysosomes (*Levitte et al., 2016*). In addition, pathogenic mycobacteria reside within specialized host structures called granulomas, a unique niche in which bacterial populations are recalcitrant to killing due to changes in bacterial physiology as well as reduced immune cell and antibiotic access (*Dartois, 2014*). Most broad screens for host-directed therapies have been limited to cell culture. While providing insight into cell-autonomous activities of potential host-directed drugs, cell culture platforms lack the multicellular interactions and complex environment present during mycobacterial disease in vivo.

Zebrafish are natural hosts of *Mycobacterium marinum*, one of the closest relatives of the *Mtb* complex (*Tobin and Ramakrishnan, 2008*). *M. marinum* and *Mtb* share conserved virulence loci and induce similar host immune responses and pathology, including the formation of granulomas (*Davis et al., 2002*). Genes identified in zebrafish as determinants of mycobacterial disease progression have also been associated with disease severity and outcome in humans (*Thuong et al., 2017*; *Tobin et al., 2012*). Because zebrafish larvae are optically transparent, both pathogen and host processes can be imaged in vivo in a natural host. *M. marinum* infection of larval zebrafish recapitulates key aspects of tuberculosis pathogenesis (*Davis et al., 2002*). In addition, the zebrafish larva's small size and relative permeability to small molecules allow for chemical genetic screening in the context of whole animals (*MacRae and Peterson, 2015*). Here, we used the zebrafish-*M. marinum* model to perform a chemical screen in whole animals for host-directed therapies during mycobacterial infection.

We identify a number of host-directed therapies with in vivo efficacy against mycobacterial infection. Among these is clemastine, a well-tolerated, broadly available, FDA-approved drug. We report that, in zebrafish, clemastine acts in vivo to enhance mycobacterial control via potentiation of the purinergic receptor P2rx7. Using light-sheet microscopy, we find that clemastine-induced potentiation of P2rx7 drives increased calcium transients within infected macrophages in vivo. P2rx7 potentiation enhances inflammasome activation, resulting in restriction of mycobacterial growth in zebrafish larvae. Finally, using a novel granuloma explant model, we show that clemastine is an effective host-directed therapy in the context of granulomas from established mycobacterial infections.

## Results

### An in vivo chemical screen uncovers six novel HDTs for mycobacterial infection

To identify novel host-directed therapies that reduce mycobacterial burden in vivo, we screened the 1200 FDA-approved drugs of the Prestwick Chemical Library using zebrafish larvae infected with fluorescent *M. marinum* (*Figure 1A*). Three animals per well in a 96-well format were each infected with ~ 150 CFU of *M. marinum* and maintained for 5 days at a 5 μM concentration of each compound. We evaluated the efficacy of each of the 1200 compounds in reducing bacterial growth by measuring mycobacterial fluorescence in whole animals over the course of a 5-day infection. Quantitation of bacterial fluorescence driven by a stable, constitutive promoter provides a validated measure of bacterial burden in zebrafish larvae (*Adams et al., 2011*; *Takaki et al., 2013*; *Walton et al., 2018*).

From the initial screen, we identified 51 compounds that reduced mycobacterial burden in vivo. In a secondary screen of these potential hits, we found that 23 of the 51 compounds identified resulted in reduced bacterial burdens in a second set of in vivo infections, and these 23 compounds were selected for further analysis. To distinguish between drugs that directly target bacteria and those that have host-dependent effects, we treated *M. marinum* cultures with each compound and monitored bacterial growth (*Figure 1A*). Three compounds, each previously reported to have antimicrobial activity, exhibited direct anti-mycobacterial activity at the concentrations tested (*Figure 1*, *Figure 1—figure supplement 1A–D*), validating our ability to recapitulate the effect of known anti-

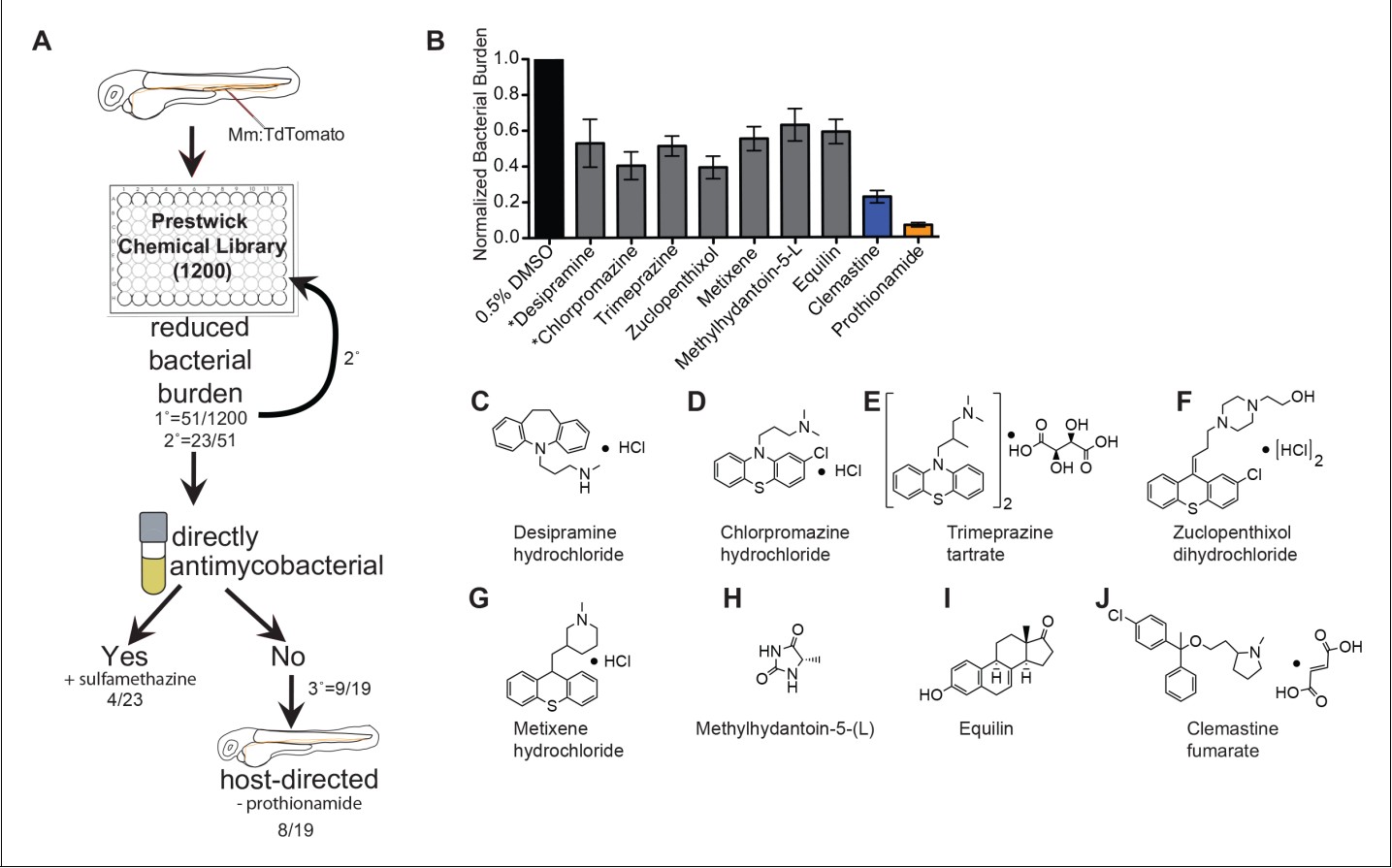

**Figure 1.** An in vivo zebrafish chemical screen identifies six novel host-directed therapies. (**A**) Schematic of chemical screen. 1200 compounds in the Prestwick Library were screened at a final concentration of 5 µM in 0.5% DMSO for substantial reductions in bacterial burden. Three larvae were placed in each well and imaged at 5 days post-infection (dpi). 1° denotes primary screen, 2° secondary, and 3° tertiary screens. Numerator represents number of hits after that screening step over the total number of compounds tested in that step. *Mm = Mycobacterium marinum* (**B**) Hits from chemical screen determined to reduce bacterial burden compared to DMSO control. All drug concentrations are 5 µM in 0.5% DMSO. Bacterial burden by fluorescence is normalized to DMSO control for each experiment. Data shown is representative of three experiments with n > 15 animals for each experiment. Error bars are s.d. *represents drugs with previously identified host-directed mechanism (**C–J**), Chemical structures of drugs identified as reducing bacterial burden in a host-directed manner.

DOI: https://doi.org/10.7554/eLife.39123.002

The following figure supplement is available for figure 1:

**Figure supplement 1.** An in vivo zebrafish chemical screen identifies four directly antimycobacterial compounds.

DOI: https://doi.org/10.7554/eLife.39123.003

tubercular drugs in an in vivo model. A known antibiotic, sulfamethazine, was not effective at 5 µM in broth culture, but was effective in vivo, suggesting that in vivo distribution kinetics and/or metabolism may enhance potency. Thus, the whole animal drug screening approach was able to identify antibiotics that are effective in vivo but might be missed in screens of axenic cultures (*Supplementary file 1*). Due to the known antimicrobial function of sulfamethazine, we did not consider it further as a potential host-directed therapy. The Prestwick Chemical Library contains additional antimycobacterial agents that did not emerge as hits; however, a number of these compounds are ineffective in zebrafish larvae at the 5 µM concentration used during our screen (*Adams et al., 2011*; *Takaki et al., 2013*).

Because host-directed therapies may provide new approaches to TB treatment, we focused on the subset of drugs with host-dependent activity. The 19 compounds that reduced bacterial burden in vivo but not in culture were obtained from independent sources and tested in larger cohorts. Of these 19 potential host-directed compounds, 9 resulted in consistently reduced bacterial burdens in

three independent in vivo experiments but had no effect on bacterial growth in liquid broth culture (*Figure 1B* and *Figure 1—figure supplement 1*). Thus, these nine drugs met all criteria for host-dependent therapies effective in reducing burden in whole animals over 5 days of infection. One of the nine, prothionamide, appeared to be host-dependent at the concentration tested but is likely not host-directed (*Figure 1B* and *Figure 1—figure supplement 1*); prothionamide is a pro-drug that can be metabolized into active anti-mycobacterial forms by both bacteria and host (*Nishida and Ortiz de Montellano, 2011*).

Of the remaining eight compounds we identified as likely to be host-directed therapies (*Figure 1C–J*), one drug, desipramine, had been described previously as a potential HDT; it is proposed to act through inhibition of acid sphingomyelinase, which has been implicated in necroptosis and the response to mycobacterial infection in zebrafish (*Roca and Ramakrishnan, 2013*). Chlorpromazine is known to inhibit acid sphingomyelinase, although it also has reported anti-mycobacterial activity in culture (*Amaral et al., 2007*). To the best of our knowledge, none of the six remaining compounds had previously been considered as a potential host-directed therapy for mycobacterial infection. We chose to focus on the most potent of these, clemastine, an inexpensive, widely available, over-the-counter antihistamine.

## Clemastine is a host-directed compound that requires host macrophages to reduce bacterial growth

We found that clemastine consistently reduced mycobacterial growth in vivo over the course of a 5-day infection (*Figure 2A–B*). The effect was dose dependent (*Figure 2C*), and at a concentration of 5 µM resulted in a ~ 60% reduction in bacterial burden by 5 days post-infection (dpi). At 5 dpi, there were fewer total infection foci, and areas of infection were smaller (*Figure 2B*), indicating that clemastine reduces bacterial burden significantly after just a few days of infection. In culture conditions, clemastine had no effect on bacterial growth at concentrations up to 50 µM, although at higher concentrations that were toxic to larvae (100 µM), there was some reduction in bacterial growth in culture (*Figure 2D*). Thus, clemastine functions to reduce bacterial burden in whole animals in a host-dependent manner.

Clemastine's mechanism of action was unlikely to be attributable to its antihistamine activity. Of the 41 antihistamines in the Prestwick Chemical Library, only one other in a separate class, trimeprazine, was identified as a hit. In addition, diphenhydramine, another antihistamine of the same family as clemastine, had no effect on bacterial burden (*Figure 2—figure supplement 1A*). Thus, we hypothesized that clemastine's host-directed antimycobacterial activity was due to a distinct effect of the drug unrelated to its antihistamine activity.

Macrophages are central host cells in mycobacterial infection in humans and animal models, including zebrafish (*Clay et al., 2007*; *Philips and Ernst, 2012*). To test whether clemastine acts via macrophages to reduce bacterial burden, we treated infected, macrophage-deficient animals with clemastine. Mutations in the gene encoding the myeloid transcription factor Irf8 result in an absence of larval macrophages, due to impaired differentiation from myeloid precursors (*Shiau et al., 2015*). In *irf8* mutants, similar to other macrophage-deficient animals, bacterial burdens are higher, with extracellular bacterial growth (*Pagán et al., 2015*). We found that clemastine was ineffective in *irf8*-deficient animals (*Figure 2E–F*), suggesting that clemastine's action requires functional macrophages.

We next took advantage of the ability to perform high-resolution, longitudinal imaging in the zebrafish to understand how clemastine alters mycobacterial infections within macrophages in vivo. The zebrafish infection model enables quantitation of intracellular growth in vivo through direct enumeration of bacterial burden and replication in individual macrophages (*Takaki et al., 2013*). Host mutations that compromise the ability of macrophages to restrict intracellular mycobacterial growth result in higher numbers of bacteria per macrophage, while bacterial mutants attenuated for intracellular growth exhibit reduced bacterial numbers within each macrophage (*Takaki et al., 2013*). We used a cerulean fluorescent *M. marinum* strain to infect the zebrafish transgenic line *Tg(mfap4:tdTomato)$^{xt12}$* (*Walton et al., 2015*), in which macrophages are fluorescently labeled red, and quantified intracellular bacterial growth during clemastine treatment (*Figure 2—figure supplement 1B*). Clemastine treatment for 24 hours reduced the number of bacteria per macrophage by ~ 30%, suggesting that clemastine enhances control of mycobacterial infection at the level of individual infected

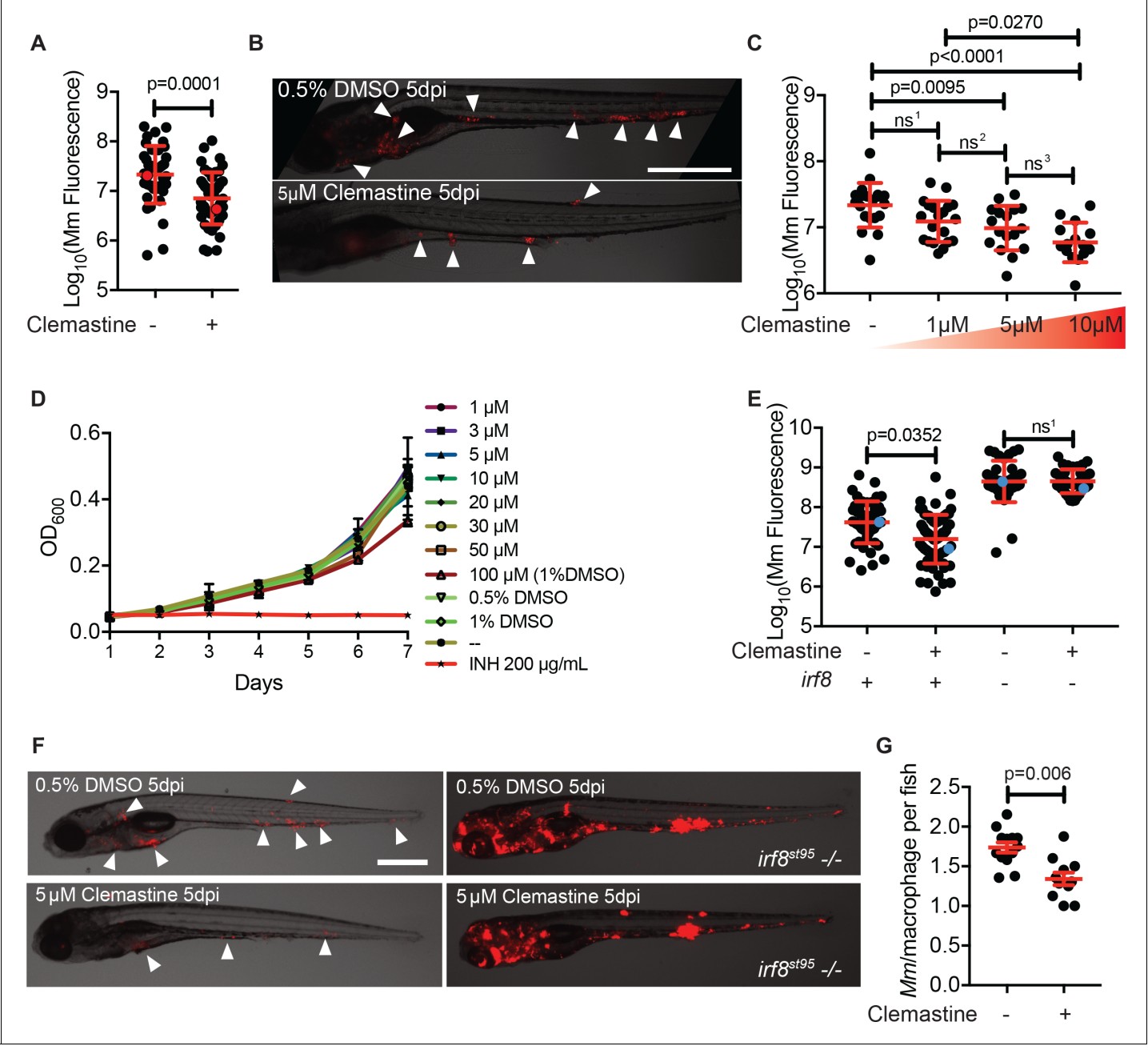

**Figure 2.** Clemastine activity is host-dependent, dose-responsive, and requires macrophages. (A) Bacterial burden per animal assessed by *Mm:tdTomato* fluorescence 5 days post infection (dpi) after treatment with 0.5% DMSO vehicle or 5 μM clemastine. Red dots denote the corresponding image shown in *Figure 2B*. Representative of five experiments. (B) Representative images from experiment in *Figure 2A* (red dots). *Mm:tdTomato* infection treated with 0.5% DMSO vehicle or 5 μM clemastine in 0.5% DMSO. White arrows denote regions of bacterial foci. Scale bar is 500 μm. (C) Quantification of *Mm:tdTomato* fluorescence at 5 dpi in zebrafish larvae treated with increasing concentrations of clemastine. (D) *Mm:tdTomato* bacterial broth culture grown in the presence of increasing concentrations of clemastine, compared to 0.5% DMSO, 1% DMSO, no vehicle (-), and 200 μg/mL isoniazid. (E) Bacterial burden of wildtype and heterozygous siblings or *irf8st95* mutants, which lack macrophages, treated with 0.5% DMSO or 5 μM clemastine. Blue dots indicate animals in *Figure 2F*. Data are pooled from two biological replicates. (F) Representative infections from *irf8st95* mutants and wildtype/heterozygous siblings from each treatment group in *Figure 2E* at 5 dpi. Wildtype animals (left) were equally brightened (no change in gamma settings) to show contrast of bacteria and brightfield. The *irf8st95*-/- example animals (right) were not brightened. (G) Number of bacteria per macrophage during treatment with 0.5% DMSO or 5 μM clemastine, 1 dpi. Each dot represents the mean number of intracellular *Mm:mCerulean* bacteria inside macrophages of one *Tg (mfap4:tdTomato)xt11* animal at 24 hpi infected with ~ 50 CFU. Representative of three independent experiments. (A) Two-tailed, unpaired t-test. (C) Ordinary one-way ANOVA with Tukey's multiple comparison test, ns[1] = 0.0864, ns[2] = 0.7799, ns[3] = 0.2415. Post-test for linear trend, p<0.0001. (E) Kruskal-Wallis ANOVA for unequal variances with Dunn's multiple comparisons test, ns[1] > 0.9999

*Figure 2 continued on next page*

*Figure 2 continued*

(A,C,E) Error bars are s.d. (G) Two-tailed unpaired t-test; error bars are s.e.m. p-Values from statistical tests on untransformed data are provided in *Supplementary file 2*.

DOI: https://doi.org/10.7554/eLife.39123.004

The following figure supplement is available for figure 2:

**Figure supplement 1.** Macrophage function during clemastine treatment.

DOI: https://doi.org/10.7554/eLife.39123.005

macrophages, even before formation of granulomas (*Figure 2G*). Importantly, clemastine did not alter the total number of macrophages per animal (*Figure 2—figure supplement 1C*).

To distinguish between growth restriction and a microbicidal effect, we performed long-term live imaging, focusing on individual macrophages within infected animals. Clemastine-treated animals consistently showed loss of bacterial fluorescence within macrophages, while control-treated animals did not, suggesting that clemastine enhances the microbicidal activity of macrophages (*Video 1*, *Figure 2—figure supplement 1D*). Together, these findings are consistent with a macrophage-dependent mechanism in which macrophage-induced mycobacterial killing is enhanced upon clemastine administration.

Pathogenic mycobacteria can limit acidification of the phagosome, creating a replicative niche within macrophages (*Rohde et al., 2007*). We sought to determine if clemastine increased the acidification of mycobacteria-containing phagosomes. The reporter *aprA′::gfp* has been used to probe acidification in mycobacteria (*Abramovitch et al., 2011*). The promoter of *aprA* is pH-inducible and drives expression of GFP only under acidic conditions (pH <6). Simultaneous visualization of the inducible GFP and mCherry expression from the constitutive promoter *smyc* (*smyc′::mCherry*), allows visualization of the acidification state of bacteria in vivo. We measured the ratio of green fluorescence to red fluorescence in *M. marinum aprA′::GFP,smyc′::mCherry* infections in vivo and observed that clemastine did not increase the proportion of bacteria that induce the *aprA* reporter (*Figure 2—figure supplement 1E*), indicating that clemastine likely does not act through increased rates of phagosome acidification.

## Clemastine enhances calcium transients in infected macrophages, and its activity is dependent on P2RX7

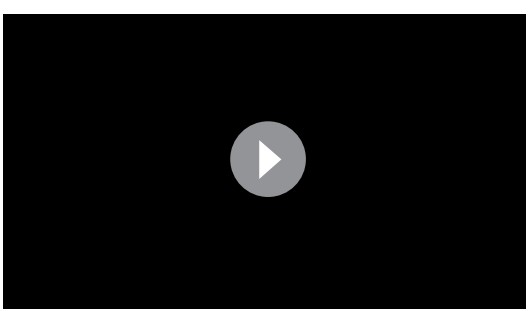

**Video 1.** Clemastine treatment results in clearing of bacteria by host macrophages. Split screen video of the caudal hematopoietic tissue of a *Tg(mfap4: tdTomato-CAAX)^xt6^* (false colored magenta) zebrafish larva infected with ~50 CFU of *Mm:mCerulean.* Left side of the screen is an animal treated with 0.5% DMSO. Right side is an animal treated with 5 μM clemastine. In vivo timelapse is 17–34 hr post infection, with frames every 10 min. Maximum intensity projection of 4 z stacks of 15 μm, 30 frames per second. Stills of single channels are in *Figure 2—figure supplement 1D*.

DOI: https://doi.org/10.7554/eLife.39123.006

P2RX7 is a ligand-gated cation channel activated by extracellular ATP, which is found at sites of infection and injury (*Di Virgilio et al., 2017*). In humans, mice, and fish, P2RX7 is highly expressed on immune cells (*Di Virgilio et al., 2017*; *He et al., 2013*). One cell culture study had reported that clemastine can potentiate human P2X7 receptor (P2RX7) activity (*Nörenberg et al., 2011*). We hypothesized that, in vivo, clemastine's effect on mycobacterial infection burden might involve potentiation of P2RX7.

Using CRISPR/Cas9-based targeting, we isolated and outcrossed two different alleles predicted to abolish the second transmembrane domain of zebrafish P2RX7 receptor (P2rx7) (*Figure 3A–B*). Both alleles (*p2rx7^xt26^* and *p2rx7^xt28^*) result in frameshifts in exon 10 and predicted stop codons upstream of the second transmembrane domain, suggesting that both alleles are functional nulls (*Smart et al., 2003*). We then examined whether these mutations affected responsiveness to clemastine treatment.

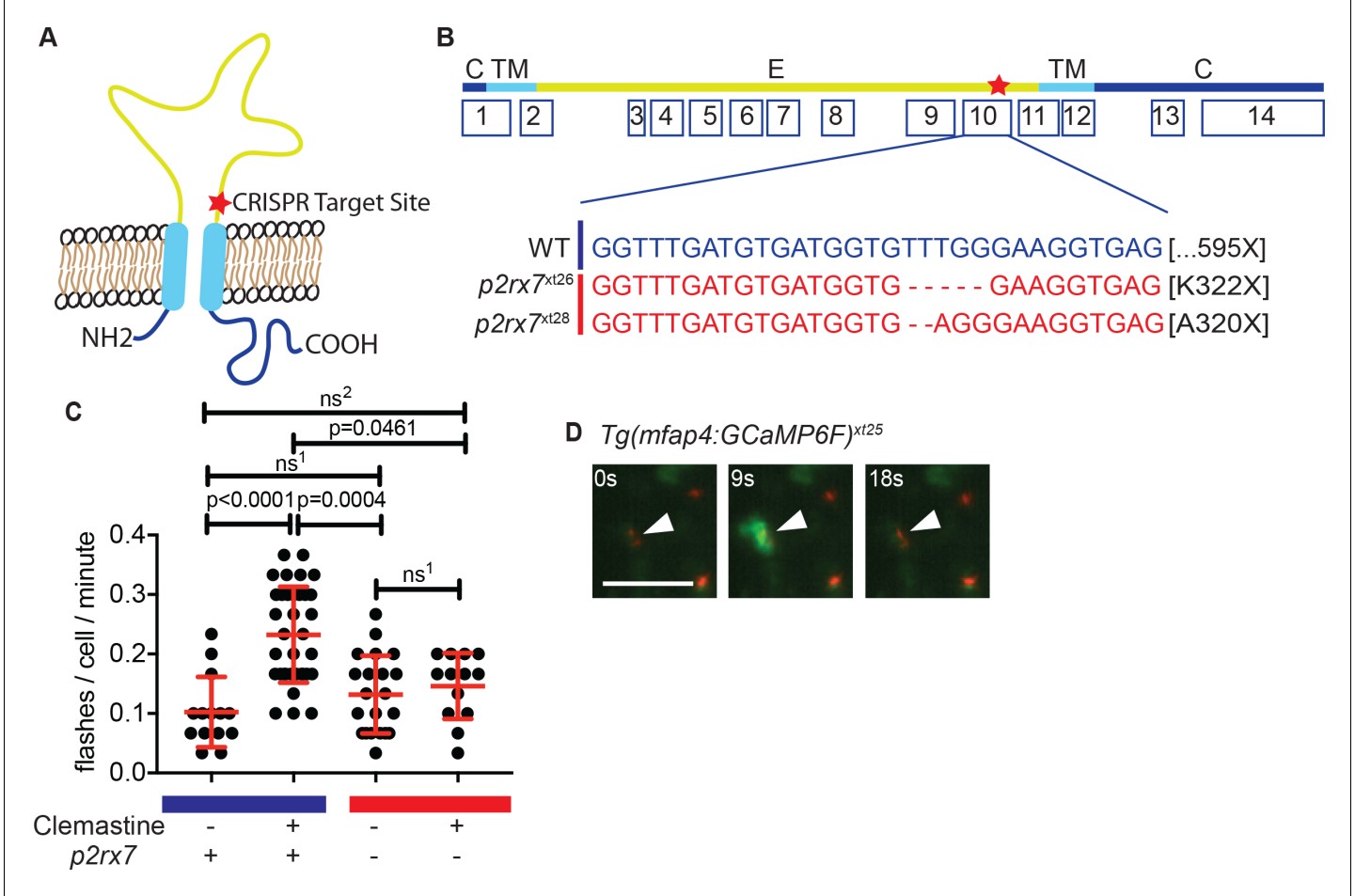

**Figure 3.** Clemastine increases frequency of macrophage calcium transients in a *p2rx7*-dependent manner. (**A**) Schematic of P2rx7 receptor, based on human structure, with CRISPR target site in exon 10 denoted with a star. (**B**) CRISPR/Cas9-mediated lesions in *p2rx7* include a five base pair deletion (*p2rx7^{xt26}*) and a two base pair deletion (*p2rx7^{xt28}*) leading to a premature stop codon in exon 10. Red bar denotes *p2rx7* mutants and blue bar denotes WT animals. (**C**) Quantification of calcium flashes observed with light-sheet microscopy in *Tg(mfap4:GCaMP6F)^{xt25}* and *p2rx7^{xt26};Tg(mfap4: GCaMP6F)^{xt25}* infected with ~ 50 CFU *Mm:tdTomato*. Each dot represents the number of flashes observed in a single infected macrophage in an animal during a 30-min time-course during treatment with 0.5% DMSO or 5 μM clemastine, 4–5 hpi, n > 3 fish for each group. All cells counted were visibly infected with *Mm:tdTomato* and scoring performed blind to genotype or treatment. (**D**) Representative calcium transient. Panels are stills from a light-sheet video in an untreated *Tg(mfap4:GCaMP6F)^{xt25}* animal infected with *Mm:tdTomato*. Scale bar is 25 μm. (**C**) Kruskal-Wallis ANOVA for unequal variances, Dunn's multiple comparison test. ns[1] > 0.9999, ns[2] = 0.7007 All error bars are s.d.; p values from statistical tests on untransformed data are provided in **Supplementary file 2**.

DOI: https://doi.org/10.7554/eLife.39123.007

The following figure supplement is available for figure 3:

**Figure supplement 1.** Light-sheet video quantification reveals calcium dynamics in macrophages.

DOI: https://doi.org/10.7554/eLife.39123.008

Given P2RX7's known role as a calcium channel, we first assessed intracellular calcium dynamics within macrophages in vivo during clemastine treatment.

To examine clemastine's effect on macrophage calcium dynamics in vivo, we generated the zebrafish transgenic line *Tg(mfap4:GCaMP6F)^{xt25}* in which the genetically encoded calcium indicator GCaMP6F is driven by a macrophage-specific promoter (*Chen et al., 2013*; *Walton et al., 2015*). Using light-sheet microscopy, we were able to examine macrophage calcium dynamics in whole animals in vivo during infection. Although calcium dynamics during mycobacterial infection have previously been studied in cultured macrophages, they have not been assessed in vertebrate models in vivo. We found that both infected and uninfected macrophages in the caudal area underwent stereotypical calcium flashes (*Figure 3C–D*) at ~ 0.1 flashes per minute (*Figure 3—figure supplement 1A*,

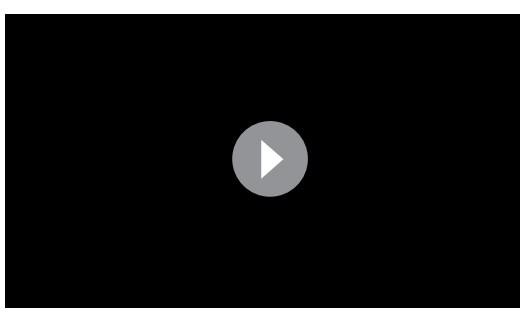

**Video 2.** Wildtype and *p2rx7* mutants exhibit similar calcium transients in uninfected larvae. Split screen video of *Tg(mfap4:GCaMP6F)^{xt25}* (left) and *p2rx7^{xt26};Tg (mfap4:GCaMP6F)^{xt25}* (right) uninfected larvae, 2 days post fertilization (dpf). 30 min light-sheet microscopy timelapse, acquiring every 8.8 s. Maximum intensity projection of 80 steps of 1 µm, 30 frames per second. Flashes are marked with either a circle (WT) or a square frame (mutant). Yellow frames represent cells that only flash once. Other colors (white, green, blue, red, green, cyan) represent cells that flash more than once, with the same cell marked in the same color throughout the timelapse. Only cells that are present during the whole video are marked. Whole cell flashes, not subcellular flickers, are marked.
DOI: https://doi.org/10.7554/eLife.39123.009

*Video 2*). Administration of clemastine significantly enhanced the frequency of these macrophage calcium transients more than two-fold in infected cells (*Figure 3C* and *Video 3*).

Given the known role of P2RX7 in mediating calcium influx, we next tested whether the enhancement of calcium transients we observed with clemastine in vivo was dependent on P2r×x7. We crossed the macrophage calcium reporter line into the *p2rx7^{xt26}* background (*p2rx7^{xt26};Tg(mfap4:GCaMP6F)^{xt25}*) and quantified calcium transients in macrophages within whole animals by light-sheet microscopy (*Videos 2–4*); we observed no differences in calcium dynamics between wildtype and *p2rx7* mutant fish at baseline (*Figure 3—figure supplement 1B*, *Video 2*). In infected wildtype animals, clemastine increased the frequency of calcium transients more than two-fold, but this effect was abrogated in *p2rx7* mutants (*Figure 3C*). Clemastine did not significantly increase calcium transients in uninfected cells in infected fish (*Figure 3—figure supplement 1C*). Thus, clemastine's enhancement of calcium transients in *Mycobacterium*-infected cells may occur through potentiation, rather than direct agonism of P2rx7.

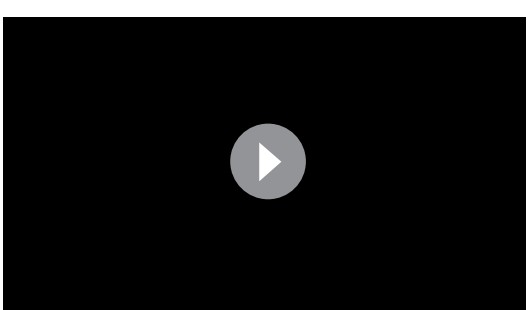

**Video 3.** Clemastine enhances calcium transients in wildtype infected zebrafish larvae. Split screen video of *Tg(mfap4:GCaMP6F)^{xt25}* larval zebrafish 2 dpf, infected with ~ 50 CFU. *Mm:TdTomato*, 4 hr post infection, treated with 0.5% DMSO (left) or 5 µM clemastine (right). 30 min light-sheet microscopy timelapse, acquiring every 8.8 s. Maximum intensity projection of 80 steps of 1 µm, 30 frames per second. Flashes are marked with either a circle or a square frame. Yellow frames represent cells that only flash once. Other colors (white, green, blue, red, green, cyan) represent cells that flash more than once, with the same cell marked in the same color throughout the timelapse. Only cells that are present during the whole video are marked. Whole cell flashes, not subcellular flickers, are marked.
DOI: https://doi.org/10.7554/eLife.39123.010

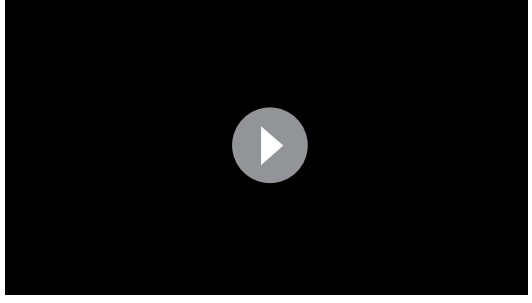

**Video 4.** Clemastine does not enhance calcium transients in *p2rx7* mutant infected zebrafish larvae. Split screen video of *p2rx7^{xt26};Tg(mfap4:GCaMP6F)^{xt25}* larval zebrafish two dpf, infected with ~ 50 CFU *Mm: TdTomato*, 4 hr post infection, treated with 0.5% DMSO (left) or 5 µM clemastine (right). 30 min light-sheet microscopy timelapse, acquiring every 8.8 s. Maximum intensity projection of 80 steps of 1 µm, 30 frames per second. Flashes are marked with either a circle or a square frame. Yellow frames represent cells that only flash once. Other colors (white, green, blue, red, green, cyan) represent cells that flash more than once, with the same cell marked in the same color throughout the timelapse. Only cells that are present during the whole video are marked. Whole cell flashes, not subcellular flickers, are marked.
DOI: https://doi.org/10.7554/eLife.39123.011

## Clemastine's anti-mycobacterial activity is dependent on *p2rx7*

Having established a dependence of clemastine-enhanced calcium flux on *p2rx7*, we next sought to determine if the accompanying reduction in bacterial burden was also *p2rx7* dependent. Using *p2rx7*[xt26] and *p2rx7*[xt28] mutant larvae and wildtype siblings, we tested clemastine's effect on bacterial burden over the course of a 5-day infection. *p2rx7* mutants did not show significant differences in overall bacterial burden, although there was a slight decrease in bacterial burden in *p2rx7* mutants (*Figure 4A*). Clemastine consistently reduced burden in wildtype animals but failed to reduce bacterial burden in *p2rx7* mutants (*Figure 4A–B*, *Figure 4—figure supplement 1A–B*). Thus, both the altered macrophage calcium dynamics and reduction in bacterial burden were dependent on functional P2rx7.

Quantitation of bacterial fluorescence has been validated as an accurate measure of bacterial burden in the zebrafish larval model (*Adams et al., 2011*; *Takaki et al., 2013*; *Walton et al., 2018*). However, we also analyzed burden using an independent assay. We quantitated mycobacterial 16S rRNA in a separate set of experiments and obtained similar results. Here, qRT-PCR revealed a ~ 0.5 $\log_{10}$ decrease with no effect in *p2rx7*[xt26] knockout animals (*Figure 4—figure supplement 1C*).

## Clemastine activity requires cytosolic sensing and a functional inflammasome

As a key second messenger, calcium is required for a variety of in vivo processes that may lead to increased microbicidal activity, including endosomal trafficking (*Fairbairn et al., 2001*) and inflammasome activation (*Ferrari et al., 2006*; *Murakami et al., 2012*). Our previous findings on unaltered acidification of mycobacteria in clemastine-treated animals (*Figure 2—figure supplement 1E*) led us to next consider inflammasome activation as a possible mechanism.

P2RX7 signaling has been shown to promote inflammasome activation (*Fairbairn et al., 2001*; *Piccini et al., 2008*), which can result in killing of intracellular pathogens (*Ferrari et al., 2006*; *Franceschini et al., 2015*; *Moreira-Souza et al., 2017*). Activation of inflammasomes is dependent on cytosolic sensing of PAMPs or DAMPs (*Lamkanfi and Dixit, 2014*). Pathogenic mycobacteria secrete a variety of effector proteins that access the host cytosol, presumably through permeabilization of the phagosomal membrane; cytosolic access is dependent on the specialized secretion system ESX-1 (*Conrad et al., 2017*; *Koo et al., 2008*; *Manzanillo et al., 2012*; *Wassermann et al., 2015*). Because P2RX7 signaling has been closely linked to inflammasome activation, we first sought to investigate if cytosolic access is required for clemastine's effect.

Pathogenic mycobacteria have been reported to both activate and limit inflammasome activation, depending on the infection model used (*Briken et al., 2013*). Mycobacteria lacking the RD1 region, which encompasses the ESX-1 type VII secretion system, fail to engage cytosol-based host responses (*Abdallah et al., 2011*; *Dorhoi et al., 2012*; *Volkman et al., 2004*). We infected zebrafish with a *M. marinum* strain lacking the RD1 region (ΔRD1) (*Volkman et al., 2004*) and asked whether clemastine retained efficacy in the absence of engagement with the host cytosol. In contrast to its effect on wildtype *M. marinum*, clemastine had no effect on burden in the ΔRD1 mutants (*Figure 5A* and *Figure 5—figure supplement 1A*).

As expected, *M. marinum* ΔRD1 mutants were attenuated during infection, and displayed reduced burden at later timepoints despite equivalent inocula. We therefore wanted to rule out the explanation that clemastine's effect was merely dependent on burden. In order to test whether clemastine is effective on infections with lower bacterial load, we infected larval zebrafish with a second attenuated mycobacterial strain. We found that *M. marinum* transposon mutants in the gene *cmaA2*, which encodes a trans-cyclopropane synthetase (*Glickman et al., 2001*), were attenuated in the zebrafish model. Despite a burden comparable to the ΔRD1 mutants (*Figure 5A* and *Figure 5—figure supplement 1B*), clemastine still was able to reduce bacterial load, suggesting that the RD1-dependent effect was not merely due to differences in burden.

Reciprocally, when we increased the inoculum of ΔRD1 mutant bacteria above wildtype levels, clemastine is still ineffective (*Figure 5—figure supplement 1B*). This was true using independent measures of bacterial burden at five dpi including fluorescence (*Figure 5—figure supplement 1B*) and 16S rRNA-based measures of burden (*Figure 5—figure supplement 1C*). Additionally, we examined early control of bacterial growth in ΔRD1 infections. During the first 24 hr of infection, clemastine reduced the number of WT bacteria per macrophage (*Figure 2G* and *Figure 5—figure*

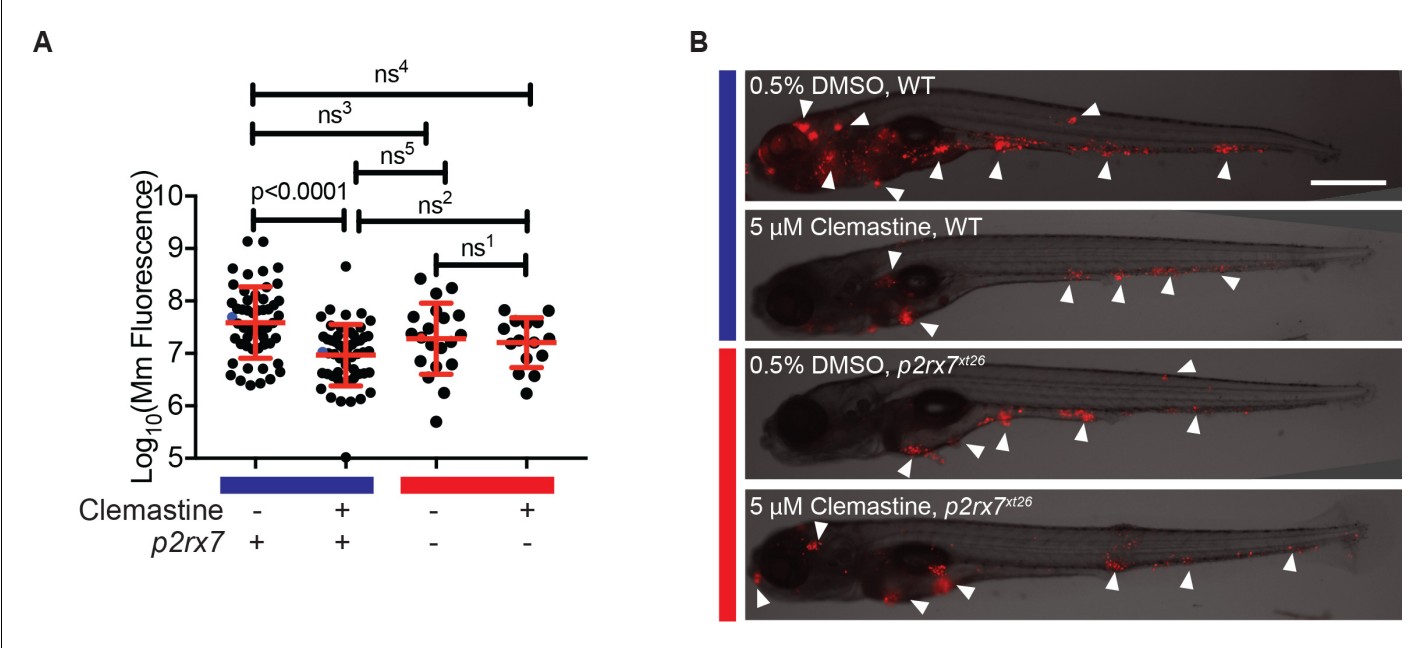

**Figure 4.** The effect of clemastine on bacterial burden is *p2rx7* dependent. (**A**) Quantification of *Mm:tdTomato* fluorescence at 5 days post-infection (dpi) in wildtype and *p2rx7^xt26* larvae treated with 0.5% DMSO or 5 μM clemastine, infected with ~ 100–150 CFU. Blue bar denotes wildtype larvae and red denotes *p2rx7^xt26* mutants. Red and blue dots denote the corresponding image shown in (**B**). Representative of 12 independent experiments, with > 15 fish per group, presented in *Figure 4—figure supplement 1A–B*. (**B**) Representative fish from each treatment group in *Figure 4A*, 5 dpi. Blue bar labels wildtype *AB fish; red bar labels *p2rx7^xt26*. Scale bar is 500 μm. White arrows mark regions of bacterial foci. (**A**) Ordinary one-way ANOVA, Tukey's multiple comparison test, ns[1] = 0.9862, ns[2] = 0.5662, ns[3] = 0.2451, ns[4] = 0.1652, ns[5] = 0.2375. All error bars are s.d.; p values from statistical tests on untransformed data are provided in *Supplementary file 2*.
DOI: https://doi.org/10.7554/eLife.39123.012

The following figure supplement is available for figure 4:

**Figure supplement 1.** Clemastine requires functional P2rx7 to reduce bacterial burden in vivo.
DOI: https://doi.org/10.7554/eLife.39123.013

supplement 1D) but failed to reduce the number of bacteria per macrophage in the ΔRD1 infection (*Figure 5—figure supplement 1D*). These data are consistent with a model in which clemastine efficacy requires a host defense pathway dependent on cytosolic engagement, regardless of bacterial load.

Given the ΔRD1 result, *p2rx7* dependence, and the links to calcium signaling and inflammasome activation, we next directly tested whether inflammasome activation might be required downstream of P2rx7 activation to promote mycobacterial restriction. We focused on the common adaptor ASC, which is required for inflammasome assembly in humans, mice, and zebrafish (*Kuri et al., 2017*; *Lamkanfi and Dixit, 2014*). We generated *asc* mutant zebrafish (zebrafish gene name *pycard*) using TALEN technology and generated a 14 base pair deletion allele, *pycard^w216*, in the first exon of the zebrafish *asc* orthologue (*Figure 5—figure supplement 2A*), resulting in a premature stop codon. As in mice, ASC-deficient animals do not exhibit dramatically increased susceptibility to mycobacterial infection (*Figure 5B*). Animals had similar burdens at five dpi, similar numbers of infection foci, and displayed no gross differences during mycobacterial infection (*Figure 5—figure supplement 2B*). However, while clemastine reduced burden in wildtype and heterozygous sibling control animals, it was completely ineffective in zebrafish *asc* mutants (*Figure 5B*, *Figure 5—figure supplement 2B–C*). These results are consistent with a model in which inflammasome activation may be limited during infection with pathogenic mycobacteria, but enhancement can lead to increased bacterial killing.

To rule out the possibility that the clemastine effect was mediated via a previously reported connection between P2RX7 and autophagy (*Mawatwal et al., 2017*), we tested the effect of clemastine on autophagy in vivo during mycobacterial infection. We made use of the zebrafish reporter line *Tg*

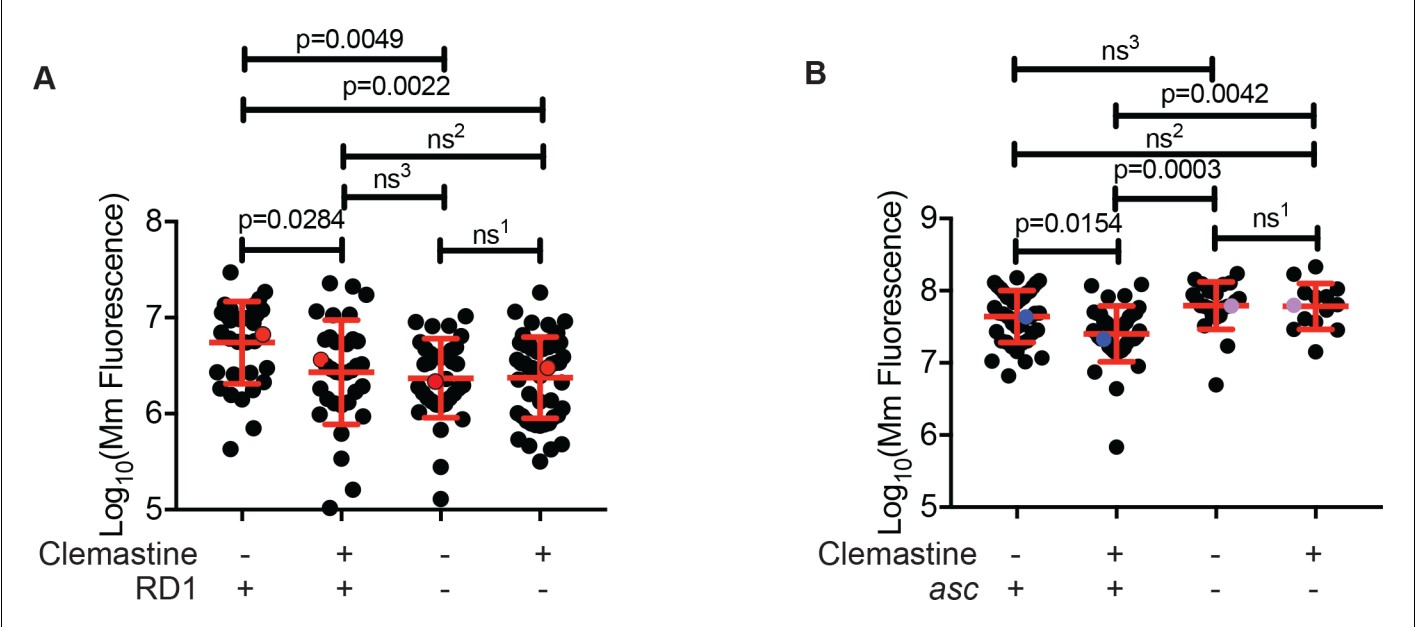

**Figure 5.** Clemastine efficacy requires cytosolic access and inflammasome signaling. (A) Quantification of bacterial burden of *Mm:Wasabi* and *Mm*ΔRD1:*Wasabi* in wildtype larvae treated with 0.5% DMSO or 5 µM clemastine, five dpi. Representative of three independent experiments. Each dot represents an individual animal's bacterial burden by fluorescence. Red dots denote the animals represented in *Figure 5—figure supplement 1A*. (B) Quantification of bacterial burden of *Mm:tdTomato* in *pycard* (*asc*) mutants and wildtype/heterozygous siblings after 0.5% DMSO or 5 µM clemastine treatment, 5dpi. Each dot represents an individual animal's bacterial burden by fluorescence. Representative of three independent experiments. Blue (WT/het) and purple (*asc* mutants) dots denote representative larvae in *Figure 5—figure supplement 2B*. Fold change over DMSO for each genotype is presented in *Figure 5—figure supplement 2C*. (A) Ordinary one-way ANOVA with Tukey's multiple comparison test. All error bars are s.d. ns[1] > 0.9999, ns[2] = 0.9452, ns[3] = 0.9430. (B) One-way ANOVA with Tukey's multiple comparison test. All error bars are s.d. ns[1] = 0.9998, ns[2] = 0.5798, ns[3] = 0.3723. p Values from statistical tests on untransformed data are provided in *Supplementary file 2*.
DOI: https://doi.org/10.7554/eLife.39123.014

The following figure supplements are available for figure 5:

**Figure supplement 1.** Clemastine does not enhance microbicidal macrophage activities in *Mm*ΔRD1 infections.
DOI: https://doi.org/10.7554/eLife.39123.015

**Figure supplement 2.** Clemastine requires inflammasome components to reduce bacterial burden.
DOI: https://doi.org/10.7554/eLife.39123.016

(*hCMV-GFP:Lc3*) (*He et al., 2009*). In this line, GFP-positive puncta are indicative of autophagosomes and can be enumerated during infection and treatment with DMSO or clemastine. Using spinning disk confocal microscopy, we observed that clemastine treatment does not enhance the number or frequency of LC3-decorated mycobacteria over the course of three days of infection (*Figure 5—figure supplement 2D*). These findings are in agreement with the observed lack of change in acidification of mycobacteria, suggesting that autophagy is not the host-directed mechanism through which clemastine reduces bacterial burden in vivo.

## Clemastine is effective in established infections

We next examined whether clemastine was an effective host-directed therapy in established infections and whether it could function therapeutically in more complex, established granulomas. Adult zebrafish granulomas share key features with human *Mtb* granulomas, including a caseous necrotic core and a tight epithelialized layer that alters bacterial physiology and may restrict access to drugs (*Cronan et al., 2016*; *Lenaerts et al., 2015*; *Swaim et al., 2006*). To directly examine clemastine's efficacy in mature, established granulomas, we made use of a novel granuloma explant model, termed Myco-GEM (*Cronan et al., 2018*). Briefly, adult animals are infected and granulomas are dissected after 2–4 weeks of infection and cultured ex vivo for a week. Because adult zebrafish are not optically transparent, this explant approach allows live visualization of granuloma and bacterial dynamics within a mature, established granuloma that contains multiple cell types. Upon multi-day

(5–7 days) treatment with clemastine, we found a nearly 70% reduction in mycobacterial fluorescence relative to vehicle control granulomas, suggesting that clemastine has efficacy both in established infections (>2 weeks post infection) and in the context of a mature granuloma (*Figure 6A–B* and *Figure 6—figure supplement 1A–B*).

Consistent with the larval experiments, we found that clemastine enhanced activation of inflammasome-dependent pathways. We used a FLICA reagent (ImmunoChemistry Technologies) that covalently binds a fluorophore to activated caspase-1; the fluorescent signal is a direct readout of caspase-1 activation. Using this tool and flow-cytometry methods developed for Myco-GEM (*Cronan et al., 2018*), we found that clemastine treatment significantly increased the percentage of cells positive for FLICA relative to the DMSO control (*Figure 6C*, *Figure 6—figure supplement 2A–C*).

To obtain an independent longitudinal readout of bacterial burden during treatment of established infections, we constructed a bioluminescent strain of *M. marinum*, *M. marinum:Lux* (*Cronan et al., 2018*). We introduced a plasmid containing a bacterial luciferase operon (*Andreu et al., 2010*) into the reference strain and infected adult animals. After 2 weeks, we dissected and cultured established granulomas, monitoring luminescence daily in the presence of clemastine or vehicle. Clemastine-treated granulomas had a nearly 10-fold reduction in luminescence relative to control granulomas after 6 days of treatment (*Figure 6D*, *Figure 6—figure supplement 3A*), suggesting that the effect size may be greater than that observed by fluorescence-based readouts. To ensure that clemastine did not quench bioluminescent signals, we treated *M. marinum:Lux* with clemastine in broth culture and saw no effect on luminescence compared to DMSO carrier (*Figure 6—figure supplement 3B*). Thus, clemastine has substantial efficacy as a host-directed therapy both in larvae and in more complex, established granulomas.

To further validate the readout of luminescence and fluorescence in the Myco-GEM model, we plated individual granulomas for colony forming units (CFU). We found that clemastine treatment significantly reduced CFU by ~ 1.4 $\log_{10}$ (~ 27 fold) compared to vehicle-exposed granulomas after a week of Myco-GEM culture (*Figure 6E*, *Figure 6—figure supplement 3A*).

Host-directed therapies will likely be used as adjunctive therapies in the context of antibiotic treatment. Therefore, HDTs should not limit the effects of antibiotics and should further reduce bacterial load. We have previously found that moxifloxacin has reduced effectiveness against mycobacteria in cultured granulomas relative to mycobacteria cultured in vitro (*Cronan et al., 2018*). Using granulomas infected with *M. marinum:Lux,* we tested clemastine's effect in the presence of moxifloxacin. Clemastine treatment significantly enhanced the effect of moxifloxacin in Myco-GEM granulomas (*Figure 6F*, *Figure 6—figure supplement 3C*), suggesting that it can act as a combination host-directed therapy with other antibiotics in the context of a complex, established infection.

## Discussion

One of the major causes of TB mortality is mycobacterial resistance to killing by standard antibiotics. Humans require multi-drug regimens and prolonged duration of treatment for active disease, increasing the risk for inadequate or discontinued treatments and the emergence of antibiotic resistance. Multi-drug resistant strains (which require 9–24 months of treatment, including use of an injectable agent) were responsible for an estimated 480,000 new TB cases in 2014, with an estimated treatment failure rate of 46% (*World Health Organization, 2018*). Thus, new treatment strategies are urgently needed.

We performed a 1200 compound chemical genetic screen of FDA-approved drugs for host-directed therapies against mycobacterial infection using a whole animal platform. Large drug screens in whole animals would be extremely challenging in other vertebrates, but the small size, relative permeability, and fecundity of zebrafish enable high-throughput chemical screens.

Using the zebrafish model, we were able to identify in vivo activity of the drug clemastine on the purinergic receptor P2rx7 during mycobacterial infection in zebrafish. We established transgenic lines to monitor calcium dynamics within macrophages during mycobacterial infection in vivo and found complex patterns of calcium transients. Interestingly, the frequency of calcium transients in zebrafish macrophages is similar to what has been reported for infected *Drosophila* macrophages in vivo (*Weavers et al., 2016*).

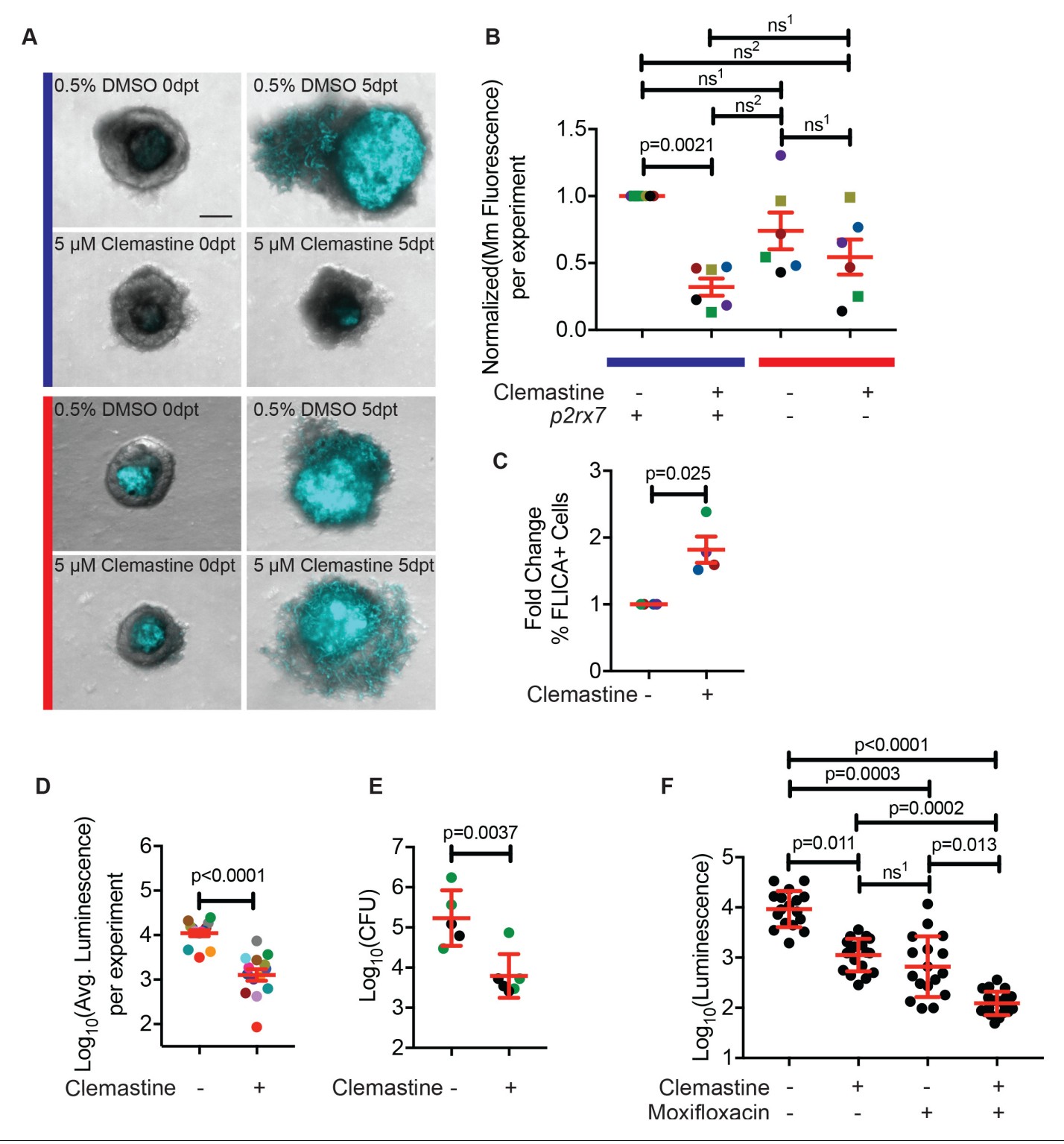

**Figure 6.** Clemastine reduces bacterial growth in granuloma explants. (**A**) Representative granuloma explants treated with 5 μM clemastine or 0.5% DMSO from wildtype or *p2rx7*[*xt26*] adult animals (2 weeks post-infection) at zero days post treatment (dpt) and 5 dpt. Animals were infected with ~300 CFU *Mm:mCerulean*. Blue line denotes granulomas from wildtype adult fish and red line denotes granulomas from *p2rx7*[*xt26*] adult fish. Images are maximum intensity projections of spinning disk confocal images, 15 steps of 10 μm. DIC and *Mm:mCerulean* channels are merged. Quantification in *Figure 6—figure supplement 1A*, averages from this experiment are denoted in the green squares of *Figure 6B*. (**B**) Quantification of 6 independent experiments in which *Mm:mCerluean* fluorescence at 5dpt is normalized to the DMSO treatment of wildtype granuloma explants. Each dot is the

*Figure 6 continued on next page*

*Figure 6 continued*

average bacterial burden at 5 dpt for each treatment group, normalized to the experiment's wildtype explants treated with DMSO. Bacterial burdens per experiment are color-coded (e.g. red dots are all from the same experiment). Squares represent granulomas from *p2rx7*xt28 adult infections and circles represent granulomas from *p2rx7*xt26 adult infections. The blue line represents wildtype animals and the red line represents *p2rx7* mutants. Fold change of bacterial burden over DMSO-treated granulomas for each genotype is in *Figure 6—figure supplement 1B* using the same colors for each experiment. (C) Quantification of FLICA-positive cells from FACS analysis after treatment with DMSO or 5 µM clemastine 2 dpt. Each dot represents the treatment averages from an independent experiment, normalized to the DMSO treatment for each experiment with at least three biological replicates per experiment. Gating strategy provided in *Figure 6—figure supplement 2A* and individual experimental averages in *Figure 6—figure supplement 2B*. (D) Quantification by luminescence of *Mm:Lux* granulomas 7 dpt after treatment with 0.5% DMSO or 5 µM clemastine, with each dot representing a single experiment's average luminescence for each treatment. Luminescence values are color-coded between experiments. Granuloma explants used for CFU plating are colored with green dots and represented in *Figure 6E* as green dots. Fold change of bacterial burden over DMSO-treated granulomas in each treatment is in *Figure 6—figure supplement 3A*. (E) Quantification of bacterial load by colony-forming units (CFU) 8 dpt with DMSO or 5 µM clemastine. Each dot represents a single granuloma. Fold change of bacterial burden over DMSO-treated granulomas in each treatment is in *Figure 6—figure supplement 3A*. (F) Quantification by luminescence of *Mm:Lux* granulomas seven dpt after treatment with 0.5% DMSO, 5 µM clemastine, 2 µg/mL moxifloxacin, or combination clemastine and moxifloxacin. Data are pooled from two experiments. Fold change of bacterial burden over DMSO-treated granulomas in each treatment is in *Figure 6—figure supplement 3C*. (B) Friedman Test (paired) ANOVA with Dunn's multiple comparisons test ns[1] > 0.9999, ns[2] = 0.1521. All error bars are s.e.m. (C) Unpaired t-test, error bars are s.e.m. (D) Paired t-test, error bars are s.e.m. (E) Unpaired t-test, error bars are s.d. (F) Kruskal-Wallis ANOVA with Dunn's multiple comparisons test, ns[1] > 0.9999. All error bars are s.d. p-Values from statistical tests on untransformed data are provided in *Supplementary file 2*.

DOI: https://doi.org/10.7554/eLife.39123.017

The following figure supplements are available for figure 6:

**Figure supplement 1.** Clemastine reduces bacterial growth in granuloma explants in a P2rx7 dependent manner.
DOI: https://doi.org/10.7554/eLife.39123.018

**Figure supplement 2.** Clemastine enhances activation of caspase-1 in granuloma explants.
DOI: https://doi.org/10.7554/eLife.39123.019

**Figure supplement 3.** Clemastine reduces bacterial growth in granuloma explants, as measured by luminescence and CFU-based assays.
DOI: https://doi.org/10.7554/eLife.39123.020

We identified an important moonlighting activity for clemastine in mycobacterial infection: potentiation of the purinergic receptor, P2RX7. That target, P2RX7, has been implicated in a variety of inflammatory and immune processes; its activation can result in bacterial killing in cell culture (*Coutinho-Silva et al., 2003*; *Fairbairn et al., 2001*; *Santos et al., 2013*) (reviewed in *Di Virgilio et al., 2017*). Mawatwal *et al* have reported that calcimycin induces P2RX7 dependent autophagic killing of *M. smegmatis* and BCG in cell culture (*Mawatwal et al., 2017*). However, we did not see enhancement of autophagy in clemastine-treated animals, suggesting that this is not clemastine's mode of action. Intriguingly, polymorphisms in the P2RX7 receptor have been associated with TB susceptibility in humans (*Li et al., 2002*; *Wu et al., 2014*) and functional changes contribute to TB pathogenesis in some cell culture models but have given divergent results in different mouse models (*Amaral et al., 2014*; *Myers et al., 2005*; *Santos et al., 2013*). In cell culture, P2RX7 activation enhances mycobacterial killing in an ATP-dependent manner (*Placido et al., 2006*), and these findings are consistent with our in vivo data with clemastine and P2RX7 activation.

Here, however, the in vivo drug screen uncovered a novel approach to modulating P2RX7 during mycobacterial infection. While previous studies have focused on the effects of loss of function of P2RX7, we show that direct potentiation of the channel enhances host control of infection. In addition, rather than constitutive activation, this mechanism avoids potential off-site detrimental effects. Potentiation of an ATP-responsive channel rather than constitutive activation thus represents a more nuanced approach to enhancing the host immune response and may avoid problems associated with excessive or inappropriate macrophage activation (*Feng et al., 2016*). Supporting the notion of a localized and infection-specific effect of the drug, only infected cells exhibited increased calcium transients during clemastine treatment.

P2RX7 activation has been strongly linked to inflammasome activation (*Di Virgilio, 2007*), but there is divergent evidence for the precise roles of inflammasome signaling during mycobacterial infection (*Briken et al., 2013*; *Mayer-Barber et al., 2010*). In mycobacterial mutants lacking ESX-1, there is a drastic reduction in IL-1B production in vitro, with effects on the AIM2 inflammasome (*Wassermann et al., 2015*). Others have shown that non-virulent mycobacteria increase inflammasome activation compared to virulent strains (*Shah et al., 2013*). We showed that clemastine is

ineffective against mycobacteria lacking the ESX-1 locus, leading us to initially explore the hypothesis that inflammasome signaling might be required for clemastine's effect in vivo.

To test this directly, we developed and assessed a zebrafish mutant that lacks the critical adaptor protein Asc. Consistent with findings in mice, the zebrafish *asc* larvae had no significant differences in mycobacterial burden (*Mayer-Barber et al., 2010*). However, mutations in other key inflammasome genes enhance susceptibility to mycobacterial infections in animal models, indicating that inflammasomes can be involved in host protection (*Fremond et al., 2007*; *Juffermans et al., 2000*). Notably, we show that clemastine is completely ineffective in inflammasome-deficient animals, supporting a model in which inflammasome signaling can be host-beneficial. Thus, inflammasome activation through P2rx7 potentiation may enhance mycobacterial killing, potentially by overcoming bacterially mediated inhibition. While excess stimulation of inflammasome signaling has also been shown to be host-detrimental (*Mishra et al., 2013*), there may be differences between a more nuanced and localized potentiation of the P2RX7 response to ATP and more general enhancement of inflammasome activation.

As FDA-approved compounds, drugs identified using this approach have the potential to move quickly into the clinic. Indeed, the most promising compound identified in our screen, clemastine, is inexpensive, widely available, and has few known safety risks. Clemastine recently showed promise in a clinical trial for patients with multiple sclerosis (*Green et al., 2017*), and – due to its safety profile – would be a prime candidate for low-risk/high-reward studies in humans.

Clemastine is effective both in early infections in zebrafish larvae and in longer term, established infections. Using a granuloma explant model, we found improved control of mycobacterial infection, suggesting therapeutic efficacy in established infections and within complex granulomas, the key host-pathogen interface in mycobacterial infections across natural host species. Taken together, these data suggest a model (*Figure 7*) in which clemastine reduces intracellular bacterial burden through potentiation of the purinergic receptor P2rx7 to induce inflammasome activation in both nascent and established mycobacterial infections. Thus, modulation of a P2RX7 inflammasome axis is a promising avenue for host-directed therapies against mycobacterial infection in established disease. Further understanding of the mechanistic basis for P2RX7-mediated mycobacterial control, as well as ways in which inflammasome activation can be modulated at discrete points in infection, may lead to new approaches to host-directed therapies.

## Materials and methods

**Key resources table**

| Reagent type (species) or resource | Designation | Source or reference | Identifiers | Additional information |
|---|---|---|---|---|
| Gene (*Mycobacterium marinum + Photorhabdus luminescens*) | Lux | Addgene, (*Andreu et al., 2010*). | ID_Addgene: pMV306hsp | |
| Strain, strain background (*Danio rerio*) | *AB | https://zfin.org/ZDB-GENO-960809-7 | | |
| Genetic reagent (*Danio rerio*) | mfap4: GCaMP6F$^{xt25}$ | this paper | | |
| Genetic reagent (*Danio rerio*) | p2rx7$^{xt26}$ | this paper | | |
| Genetic reagent (*Danio rerio*) | p2rx7$^{xt28}$ | this paper | | |
| Genetic reagent (*Danio rerio*) | asc$^{w216}$ | this paper | | |
| Genetic reagent (*Danio rerio*) | mfap4:tdTomato | *Walton et al., 2015*, https://doi.org/10.1371/journal.pone.0138949 | | |

*Continued on next page*

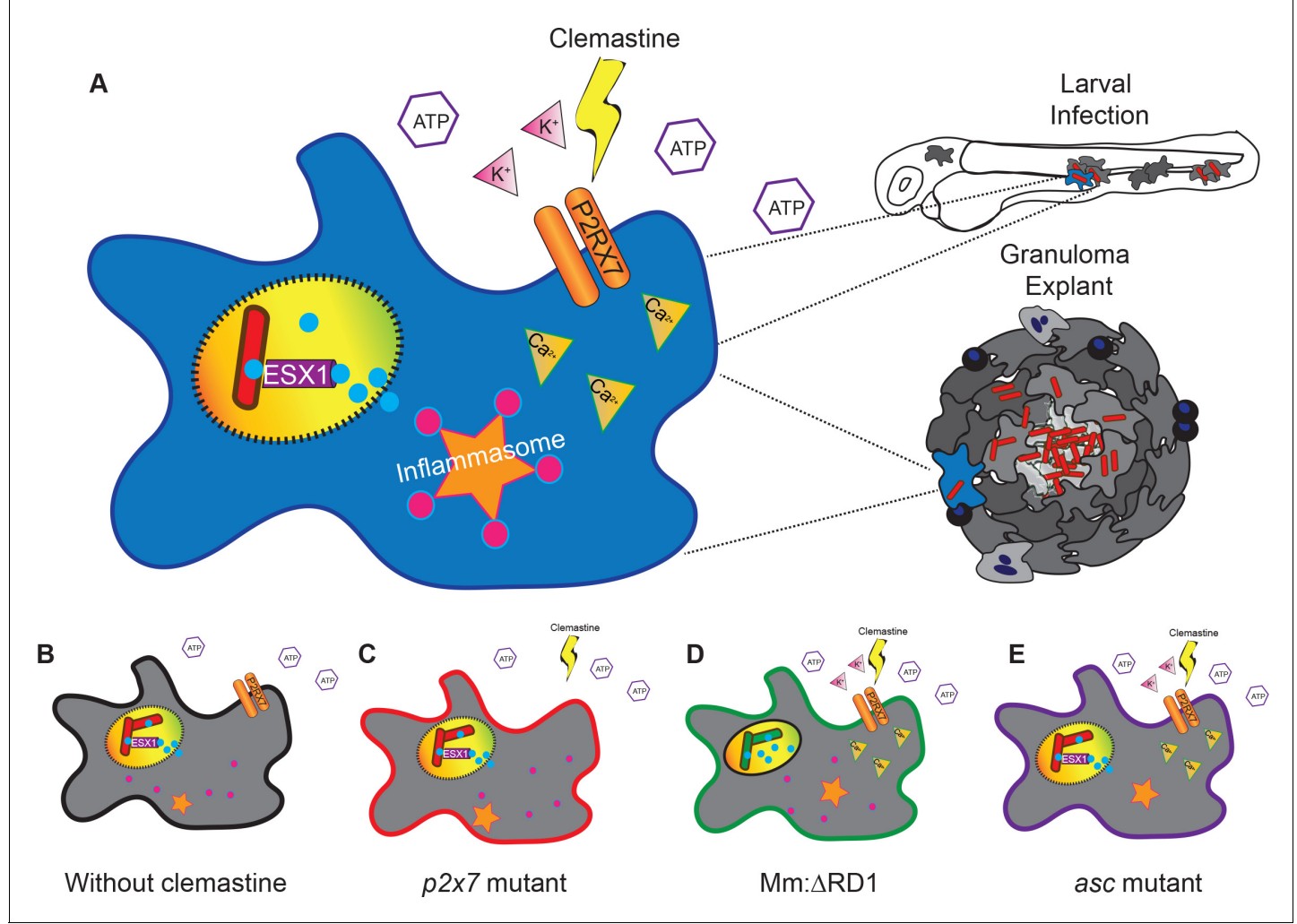

**Figure 7.** A model describing clemastine as a host-directed drug against mycobacterial infection in both nascent and established infections. Clemastine potentiates the purinergic channel, P2rx7, to increase calcium transients within infected macrophages. Clemastine enhances microbicidal activities within macrophages to kill intracellular bacteria (**A, B**) in a manner that requires *p2rx7* (**C**), intact mycobacterial ESX-1 (**D**), and a functional inflammasome (**E**).

DOI: https://doi.org/10.7554/eLife.39123.021

*Continued*

| Reagent type (species) or resource | Designation | Source or reference | Identifiers | Additional information |
|---|---|---|---|---|
| Genetic reagent (*Mycobacterium marinum*) | *cmaA2* | this paper | | |
| Genetic reagent (*Mycobacterium marinum*) | ΔRD1 | *Volkman et al., 2004*, https://doi.org/10.1371/journal.pbio.0020367 | | |
| Commercial assay or kit | FLICA | ImmunoChemistry Technologies | ID_Immuno Chemistry Technologies:9122 | |
| Chemical compound, drug | Clemastine fumarate | Sigma | ID_Sigma:SML0445 | |

*Continued on next page*

Continued

| Reagent type (species) or resource | Designation | Source or reference | Identifiers | Additional information |
|---|---|---|---|---|
| Software, algorithm other, Chemical Screening Platform | ImageJ | *Rueden et al., 2017* | | |
| Other, Chemical Screening Platform | Prestwick Chemical Library | http://www.prestwickchemical.com/libraries-screening-lib-pcl.html | | |

## Animal handling and maintenance

All zebrafish procedures and husbandry complied with policies by the Duke University Institutional Animal Care and Use Committee (protocol A122-17-05). Eggs were collected from in-tank spawning and kept in 1X E3 medium (5 mM CaCl, 178 μM KCl, 328 μM $CaCl_2$, 400 μM $MgCl_2$ in $dH_2O$) at 28.5°C. Pigmentation was prevented using 1-phenyl-2-thiourea (PTU; Alfa Aesar L06690; 45 μg/mL), beginning treatment at one dpf. Unless otherwise noted, all zebrafish are in the *AB wildtype strain.

## Transgenic lines

The transgenic line *Tg(mfap4:tdTomato)*[xt12] and *Tg(mfap4:tdTomato-CAAX)*[xt6] have been previously described (*Walton et al., 2015*). *irf8*[st95] mutant zebrafish are described elsewhere (*Shiau et al., 2015*). Creation and validation of the *tnf:gfp* line, (*TgBAC (tnfa:GFP)*[pd1028]) has been previously described (*Marjoram et al., 2015*). The transgenic line *Tg(mfap4:GCaMP6F)*[xt25] was made by injecting Tol2 transposase mRNA and *tol2*-containing DNA constructs into zebrafish embryos at the one cell stage. The constructs were assembled with Tol2kit reagents and subsequent Gateway Cloning (Invitrogen) (*Kwan et al., 2007*). The 5′ element containing the *mfap4* promoter, has been previously described (*Walton et al., 2015*). The middle element GCaMP6F was generated by PCR amplification of the *GCaMP6F* coding region from the Addgene plasmid #40755 using primers containing attB1 and attB2 sites followed by recombination into pDONR221. The 3′ element was SV40polA in pDONR P2R-P3. The destination vector utilized was pDestTol2pA.

## Generation of zebrafish lines possessing *p2rx7* loss-of-function alleles

Loss-of-function alleles of *p2rx7* were generated by targeting the sequence 5′- GGTTTGATGTGA TGGTGTTTGG-3′ in exon 10 using CRISPR/Cas9 genome editing. The gRNA in vitro transcription template was generated as described previously (*Jao et al., 2013*). Briefly, single-stranded DNA oligos 5′-GGTTTGATGTGATGGTGTT-3′ and 5′-AAACAAACACCATCACATCAA-3′ were annealed and inserted into the T7cas9sgRNA2 vector (*Jao et al., 2013*). The resulting plasmid was linearized with BamHI (New England Biolabs R0136S), purified, and 400 ng was used as a template for in vitro transcription using the T7 MEGAshortscript kit (ThermoFisher AM1354). gRNAs were co-injected with Cas9 mRNA into single cell *AB, wildtype embryos. CRISPR/Cas9-mediated mutations were determined by HRMA (described below) and Sanger sequencing. Two alleles were maintained: a five base pair deletion *p2rx7*[xt26] and a T to A transversion with a two base pair deletion, *p2rx7*[xt28]. Both mutations cause a premature stop codon in exon 10 (*Figure 3A–B*). The *p2rx7* mutant lines were crossed into *Tg(mfap4:GCaMP6F)*[xt25], *Tg(mfap4:tdTomato)*[xt12], and *Tg(mfap4:tdTomato-CAAX)*[xt6] and subsequently homozygosed in each transgenic background. Homozygous *p2rx7* mutants were viable and exhibited no apparent anatomical or fertility defects. *asc/pycard* loss-of-function alleles were generated by TALEN-mediated targeting (*Dahlem et al., 2012*) of *pycard* exon 1, resulting in a 14 base pair deletion in the PYRIN domain (*Figure 5—figure supplement 2A*). *pycard* mutant animals are viable as adults and of similar size and fecundity.

## Characterization and maintenance of *cmaA2* transposon mutant *M. marinum*

### Identification of transposon insertion sites

Two transposon mutants with disruptions in the *cmaA2* ORF, designated *cmaA2*[Tn01901] and *cmaA2*[Tn02791], were identified from a sequenced library of *M. marinum* transposon mutants (C. Cosma and L. Ramakrishnan). The previously identified insertion sites were confirmed by semi-

random PCR and sequencing, using a pool of semi-random primers in conjunction with a pair of nested primers annealing within the 3' end of the transposon. Primer sequences are as follows: Semi-random pool: 5'-GCAACNNNNGTCTCGTTAGCTCGCTGGCC-3'; 5'-ATATCNNNNGTCTCG TTAGCTCGCTGGCC-3'; and 5'-GTACTNNNNGTCTCGTTAGCTCGCTGGCC-3', where N denotes random nucleotide insertion during primer synthesis (Integrated DNA Technologies). Outer transposon-specific primer (TnMarR3): 5'-ACAACAAAGCTCTCACCAACCGTG-3'; Inner transposon-specific primer (TnMarR2): 5'-CAGACACTGCTTGTCCGATATTTGATTTAGG-3'.

The semi-random primer pool and TnMarR3 were used to perform an initial, unbiased amplification of a region around any possible transposon insertion junctions. A second amplification was then performed using a primer corresponding to the constant region of the semi-random pool (5'-GTC TCGTTAGCTCGCTGGCC-3') and TnMarR2. The product of the second amplification was sequenced using TnMarR2, and the output was aligned to the *M. marinum* genome (BLASTn, NIH). The presence of a single sequence aligning to the *cmaA2* locus confirmed the presence of only a single transposon insertion in each genome and disruption of the *cmaA2* ORF for both $cmaA2^{Tn01901}$ and $cmaA2^{Tn02791}$. The insertion site for $cmaA2^{Tn01901}$ is located ~ 14% into the ORF between the first and second bases of the 42nd codon. The insertion site for $cmaA2^{Tn01901}$ is located ~ 75% into the ORF between the second and third bases of the 230th codon. $CmaA2^{Tn01901}$ (Tn01901) was selected for further study.

## Generation of fluorescent transposon mutant strains

The mutants possess the TnMar transposon, conferring resistance to Hygromycin B. Fluorescent strains were generated by electroporating (800 Ω, 25 μF, 2.5 kV, 0.2 cm gap) msp12:mCerulean-KanR (a gift from L. Ramakrishnan, University of Cambridge) into either mutant strain. Selection on media containing Kanamycin (20 μg/ml) and Hygromycin B (50 μg/ml) yielded mutant clones expressing the mCerulean fluorescent protein.

## Bacterial preparations

Wild-type *Mycobacterium marinum* (*M. marinum*) constitutively expressing a fluorescent protein (tdTomato, mCerulean, or Wasabi) and a Hygromycin resistance cassette driven by the *msp12* promoter were grown to late log phase ($OD_{600} = 0.8$) at 33°C in liquid 7H9 complete media (Middlebrook 7H9 base, Difco BD 271310) supplemented to a final concentration of 10% OADC (50 g/L BSA, 0.5% oleic acid, 20 g/L dextrose, 8.5 g/L NaCl), 0.05% Tween-80 (Fisher BP338-500), and 50 ug/mL Hygromycin B (Sigma H0654). *M. marinum* ΔRD1 was a gift from Lalita Ramakrishnan (*Volkman et al., 2004*). Bacteria were prepared into single-cell suspensions, as described (*Takaki et al., 2013*). Single-cell suspension aliquots were diluted to approximately $1 \times 10^8$ cfu/mL in 5 μL 7H9 media + OADC and were stored at −80°C long-term and diluted in 7H9 to an appropriate concentration prior to injection.

## Bacterial broth growth assay

Wild-type *M. marinum* expressing tdTomato was grown to $OD_{600} = 0.8$ and then diluted 1:10 in 7H9 complete media supplemented with 50 ug/mL Hygromycin B in 15 mL culture tubes (Falcon). The bacterial culture was supplemented with 0.5% DMSO (MPI), 200 μg/mL Isoniazid (Sigma-Aldrich I3377), 10 μg/mL moxifloxacin (Matrix Scientific 047902) or 5 μM of each chemical and grown in a 33°C shaker for 7–8 days. Each day, 100 μL of the broth was placed in a cell culture 96-well microplate plate (GBO 655090) and analyzed with an Enspire 2300 Multilabel Plate Reader (PerkinElmer) for $OD_{600}$ and fluorescence intensity, with three technical replicates per sample. The same methods were used for a growth curve in the presence of clemastine (Sigma, SML0445) for both *M. marinum* expressing mCerulean and *M. marinum* ΔRD1 expressing tdTomato (data not shown).

## Larval zebrafish infection

Two dpf larvae were anaesthetized with tricaine (MS-222; Sigma-Aldrich 160 μg/ml) and injected with fluorescent *M. marinum*. Single-cell suspension aliquots are diluted in 7H9 with 0.5% phenol red indicator (Sigma 0290) prior to injection. Between 50 and 200 CFU of fluorescent *M. marinum* were injected using a borosilicate needle (Sutter Instruments BF100-58-10), pulled with a WPI PUL-1000 system (factor setting 0) and broken approximately 8 mm from the tip. Injections are performed

using a FemtoJet injector (Eppendorf). After infection, larvae were recovered in E3 +PTU for 4 hr at 28°C. For examination of mutant phenotypes, infections were performed blind to larval genotype.

## Drug treatments

Four hours post-infection, larvae were transferred to wells containing E3 + PTU supplemented with the compound being tested. For most studies, larvae were kept in six-well plates (COSTAR 3736) containing E3 + PTU supplemented with either 0.5% DMSO (MP Bio CAS 67-68-5) or 5 μM clemastine fumarate (Sigma-Aldrich SML0445) dissolved in 100% DMSO to a final concentration of 0.5% DMSO in each treatment group. Diphenhydramine hydrochloride (Sigma-Aldrich D3630) was dissolved in 100% DMSO and diluted to a final concentration of 5 μM. All drugs were added directly to E3 + PTU. Whenever agarose was used to mount fish, the same concentration of drug was added to the agarose prior to solidification.

## Live imaging and quantification of bacterial burden

Epifluorescence microscopy was carried out on an inverted Zeiss Observer Z1 microscope using 2.5X, 5X, or 20X objectives, depending on experiment. Prior to imaging, larvae were anaesthetized with tricaine (160 μg/ml) and arrayed on a microscope slide or embedded in 0.75% low melting point agarose (Fisher BP165) in a 35 mm petri dish (MatTek). Bacterial burden by fluorescence is calculated using ImageJ (*Rueden et al., 2017*). Images are analyzed for the mean fluorescence and area of fluorescence above a threshold within the zebrafish. The threshold is empirically determined for each experiment to ensure that every infected animal has a non-zero value while not including any background autofluorescence. The threshold is kept constant between treatments within an experiment. The product of the mean fluorescence and area is computed and presented as 'Mm Fluorescence.' Granuloma explants were visualized on a spinning disk confocal (Andor) using 10x/0.3 UPlanFl N dry, WD: 10 mm, FN26.5, UIS2 objective, acquiring images with an Andor Ixon3 897 512 EMCCD, 1.2x auxilary magnification camera. Z stacks were taken at 10–15 μm, and images are assembled as maximum intensity projections using ImageJ (*Rueden et al., 2017*). During image analysis, experimenter was blinded to the genotype of the larvae.

## Light-sheet microscopy

Light-sheet fluorescence microscopy experiments were carried out on a Zeiss Light-sheet Z.1 using a Plan-Apochromat 20X/1.0 NA Aqueous immersion objective, situated with a C.mos PCO.edge camera with 16bit 1920 × 1920 sensors. One track, emission selection: 488/498 (GFP) and 560/571 (tdTomato). Light-sheet thickness was 4.32 microns with a continuous drive, 1x zoom, dual side illumination. For these experiments, two dpf zebrafish larvae *Tg(mfap4:GCaMP6F)[xt25]* and *p2rx7[xt26];Tg (mfap4:GCaMP6F)[xt25]* were infected as described and treated with DMSO or clemastine at four hpi. One to 2 hr post-treatment, larvae were placed in 1.5% low-melting point agarose, supplemented with 80 μg/ml tricaine and either 0.5% DMSO or 5 μM clemastine fumarate (Sigma-Aldrich SML0445) in a 3 mm OD glass capillary. A dual-colored z-stack was acquired every 8.8 s, collecting 80 z-steps, using a z-step size of 1 μm for 30 min.

Light-sheet videos were projected as maximum intensities and analyzed using ImageJ (*Rueden et al., 2017*). Regions of interest (ROIs) were defined as single macrophages that expressed baseline levels of green fluorescence by GCaMP6 and flashed at least once over the course of the video. Macrophages outside of the caudal hematopoietic tissue were excluded. Only individual flashes were counted, not coordinated flashes during which every macrophage, infected and uninfected, underwent a calcium flash simultaneously. If the macrophage was not in frame for the entire 30 min, it was not included in the analysis. Quantitative analyses of macrophage calcium flashes were performed blind to the treatment type and genetic identity of the specimens under observation.

## Adult infections and granuloma explants

Adult zebrafish (>2 months old) were infected as described (*Oehlers et al., 2015*). Briefly, adult fish (>2 months old) were anesthetized by immersion in 100 μg/mL tricaine and injected intraperitoneally with 300 CFU of *Mm:mCerulean* or *Mm:tdTomato* using insulin needles (BD 08290-3284-38). Adult fish were kept in 1L tanks in a dedicated infection incubator with light/dark cycles at 28°C, with water

changes and feeding every day. Between two and four weeks post-infection, granulomas were harvested using a Myco-GEM method (*Cronan et al., 2018*). Briefly, after adult fish are euthanized with a lethal dose of tricaine, the body cavity is exposed in sterile L15 medium (Gibco 21083–027). Granulomas can be dissected out with forceps or gently removed by a glass pipette. Granulomas are serially washed in sterile L15 and size-matched for treatment groups. The bottom of an optical-bottom 96-well plate (GBO 55090) is covered with 40 µL of 5 µg/mL Matrigel Matrix (Corning 354262), in L15. Granulomas are embedded in 2.5-dimensional top media with a final concentration of 5% FBS (Sigma 2442), and 1 µg/mL Matrigel matrix (Corning 354262) in L15 (Gibco 21083–027) as described (*Cronan et al., 2018*). The top and bottom media are supplemented with either DMSO (0.5%) or clemastine (5 µM in 0.5% DMSO). Granulomas are imaged via microscopy at d0 and 5 days post-treatment (dpt) using epifluorescence microscopy on an inverted Zeiss Observer Z1 microscope and on a spinning disk confocal (Andor). *M. marinum:Lux* granulomas were imaged daily as described below. Image acquisition and analyses were performed blind to the genotype of the granulomas.

## Bioluminescence assay

For this assay, we used a bioluminescent reporter strain of *M. marinum* (*Mm*-Lux) as described in *Cronan et al. (2018)*. Adult zebrafish were infected as described above and granuloma explants cultured in the same methods as described above but were kept in white 96-well cell culture microplates (GBO 675098) and luminescence was determined using an EnSpire 2300 Multilabel Plate Reader obtained from Perkin Elmer (Waltham, MA). Drug treatments were applied in the same fashion as fluorescence assays, with drug added to both bottom media and top media.

## Flow cytometry and FLICA staining of granuloma explants

For all flow cytometry experiments, we cultured granuloma explants as described above for 48 hr in the presence of 0.5% DMSO or 5 µM clemastine. Granulomas were rinsed in sterile L15 and PBS. Following manufacturer's directions, 660-FLICA Caspase Assay (Immunochemistry Technologies ICT097) was reconstituted with 50 µL 100% DMSO to make 150X stock. This stock was aliquoted into single use tubes and kept at −20°C and not freeze-thawed more than once. Just prior to use, the 150X stock was diluted 1:5 in PBS to make 30X stock. Grouped granulomas (~10 per group) were placed in 1.5 mL tubes with 97 µL PBS and 3 µL of 30 × 660 FLICA stock. Granulomas were kept at 28°C for 1 hr. Granulomas were washed in 1X apoptosis buffer and spun at 1200xg for 1 min three times. Granulomas were resuspended in 1 mL 0.05% trypsin in 1X EDTA (ThermoFisher Scientific, 25300) and kept on a nutator for 40 min at 30°C. Granulomas were centrifuged for 5 min at 300xg and rinsed with 1 mL 5% FBS in PBS three times, gently pipetting off supernatant each time, resuspended in 300 µL 5%FBS in PBS and pipetted over a 30 µm cell strainer (MACS Miltenyi 130-098-458). Single cell suspensions were analyzed on a FACS Canto II (BD Biosciences) using an ApcCy7 laser (633 nm) excitation. Data was collected using FACSDiva Software (BD Biosciences) and analyzed using FlowJo v10 (Treestar).

## Colony-forming unit (CFU) assay on granuloma explants

Granulomas were cultured as described above for up to a week. On the final day of treatment, individual granulomas were washed in sterile PBS, as in *Cronan et al. (2018)*. Briefly, individual granulomas were homogenized in sterile 7H9 media in a microcentrifuge tube with two sterile glass beads. Homogenates were diluted incrementally to $10^{-6}$ and plated on 7H10 media (BD Difco 262710) supplemented with 20 µg/mL kanamycin (Sigma-Aldrich K0129). Plates were incubated for 12–14 days at 30°C.

## Prestwick chemical library screen

The Prestwick Chemical Library (Prestwick Chemical, Illkirch, France) was stored at −80°C at 10 mM and aliquoted manually into 1 mM working stock daughter plates in 100% DMSO. The daughter plates were stored at −20°C until used in 96-well format. Wildtype zebrafish larvae carrying a TNF reporter transgene (*Tg(BAC(tnfa:GFP)[pd1028]*) (*Marjoram et al., 2015*) were arrayed in 96-well optical plates (GBO 655090) with three infected and one uninfected fish per well, in 200 µL of E3 + PTU and 5 µM of each chemical. First and last rows of each plate were maintained as 0.5% DMSO controls. Each plate contained 80 drugs, with 15 plates in all. Burden was assayed by fluorescence area and

mean intensity of fluorescent mycobacteria at five dpi. In the secondary screen, hits were screened again in the same format (three infected fish, one uninfected fish) and drugs that continued to reduce bacterial fluorescence area and mean intensity were tested again in a tertiary screen with 20 fish per group, with DMSO-treated fish as control.

## Determination of relative *M. marinum* burden by 16S rRNA qRT-PCR

Total RNA was isolated from each group of larvae using TRIzol Reagent (Thermo Fisher Scientific 15596026). An input mass of 1 μg of RNA per group was used for cDNA synthesis using the iScript cDNA Synthesis Kit (Bio-Rad 1708890). Average relative bacterial burdens were quantified by qPCR (Power SYBR Green PCR Master Mix, Thermo Fisher Scientific 4368577) targeting the *M. marinum* 16S locus. F primer: 5'-CGATCTGCCCTGCACTTC-3'; R primer: 5'-CCACAGGACATGAATCCCGT-3'. Zebrafish β-actin transcript levels were used as the internal input control. F primer: 5'-CGAGCAG-GAGATGGGAACC-3'; R primer: 5'-CAACGGAAACGCTCATTGC-3' (*Walton et al., 2018*).

## Genotyping assays

All genotyping assays were performed on fin-clips from adult zebrafish or post-experiment on whole larval animals, using genomic DNA isolated as described (*Meeker et al., 2007*). *p2rx7xt26* and *p2rx7xt28* were genotyped via KASP assays (LGC Genomics, Middlesex UK). *pycardw216* mutant larvae were genotyped using HRMA (Applied Biosystems MeltDoctor HRM 4415440). *irf8* mutations were genotyped using PCR amplification and AvaI restriction digest, as described (*Shiau et al., 2015*). Additionally, HRMA primers were developed to more rapidly genotype *irf8* mutant fish. All initial mutations were confirmed via Sanger sequencing.

Primer sequences are as follows: *pycard* HRMA primers:
5' AATCGAAAAGCTGAAAGACGAG 3'
5' ACTTACATTGCCCTGTGTTCCT 3'
*p2rx7* sequencing primers:
5' TGCGTGCCAAACATTACACTT 3'
5' AGACGTTGTGTGGGATGTGG 3'
*irf8* HRMA primers:
5' CGGCATACTAGTGAAGTAAAGG 3'
5' CTATAAGCCACTGTTTCAGT 3'
*irf8* sequencing primers:
5' ACATAAGGCGTAGAGATTGGACG 3'
5' GAAACATAGTGCGGTCCTCATCC 3'

## Statistical analysis

All statistics were performed using Prism (GraphPad), version 7. Statistics presented are performed on data presented for each graph (e.g. if a graph displays y=log$_{10}$(y) transformation, then the p values are from a statistical test on that data). All p values from statistical tests performed on transformed and untransformed data are provided in *Supplementary files 2* and *3*.

## Acknowledgements

We are grateful to J Coers and members of the Tobin lab for helpful discussions and comments on the manuscript, E Hunt for zebrafish care, W Brewer, S Espenschied, R Finethy, C Murdoch, and A Xet-Mull for experimental advice, C Cosma and L Ramakrishnan for the transposon mutants, R Abramovitch for the *aprA* construct, C Shiau for the *irf8* mutants, and A Meijer and M Varela Álvarez for experimental input and advice. This work was supported by a National Science Foundation GRFP (MAM), NIH Shared Instrumentation Grant 1S10OD020010, an American Cancer Society Postdoctoral Fellowship (MRC), a Vallee Scholar Award, and NIH grants AI130236, AI125517, AI127115 (DMT), and NIH grants HD007233, and AI116908 (REH).

## Additional information

### Funding

| Funder | Grant reference number | Author |
|---|---|---|
| National Science Foundation | GRFP | Molly A Matty |
| American Cancer Society | Postdoctoral Fellowship | Mark R Cronan |
| National Institutes of Health | HD007233 | Rafael E Hernandez |
| National Institutes of Health | AI116908 | Rafael E Hernandez |
| National Institutes of Health | 1S10OD020010 | David M Tobin |
| National Institutes of Health | AI130236 | David M Tobin |
| National Institutes of Health | AI125517 | David M Tobin |
| National Institutes of Health | AI127115 | David M Tobin |
| Vallee Foundation | Vallee Scholar Award | David M Tobin |

The funders had no role in study design, data collection and interpretation, or the decision to submit the work for publication.

### Author contributions

Molly A Matty, Conceptualization, Formal analysis, Investigation, Methodology, Writing—original draft, Writing—review and editing; Daphne R Knudsen, Rebecca W Beerman, Formal analysis, Investigation, Writing—review and editing; Eric M Walton, Mark R Cronan, Charlie J Pyle, Investigation, Methodology, Writing—review and editing; Rafael E Hernandez, Supervision, Funding acquisition, Investigation, Methodology, Writing—original draft, Writing—review and editing; David M Tobin, Conceptualization, Resources, Formal analysis, Supervision, Funding acquisition, Writing—original draft, Writing—review and editing

### Author ORCIDs

Molly A Matty (iD) http://orcid.org/0000-0002-4542-2800
Rafael E Hernandez (iD) http://orcid.org/0000-0003-4408-7411
David M Tobin (iD) https://orcid.org/0000-0003-3465-5518

### Ethics

Animal experimentation: This study was performed in strict accordance with the recommendations in the Guide for the Care and Use of Laboratory Animals of the National Institutes of Health. All of the animals were handled according to approved institutional animal care and use committee (IACUC) protocols (#A122-17-05) of Duke University.

### Decision letter and Author response

Decision letter https://doi.org/10.7554/eLife.39123.027
Author response https://doi.org/10.7554/eLife.39123.028

## Additional files

### Supplementary files

• Supplementary file 1. Summary of identified compounds. Compounds with blue background are host directed and compounds in red are known anti-infectives. Asterisk (*) represents compounds with previously identified host-directed anti-mycobacterial activities.
DOI: https://doi.org/10.7554/eLife.39123.022

• Supplementary file 2. Summary of p values and statistical tests for *Figures 1–6*. Statistics performed on transformed data are in yellow and untransformed is in gray. The table is organized by figure number, in chronological order. Statistical tests are listed and transformation equation

provided, where applicable. When a paired t-test is performed on paired data, the results of the unpaired t-test are also given. When both paired and unpaired are potentially appropriate (in the case of paired data with ineffective pairing), both test results are given. Statistical test results in *italics* denote tests not presented within the figures.
DOI: https://doi.org/10.7554/eLife.39123.023

• Supplementary file 3. Summary of p values and statistical tests for the figure supplements to Figures 1-6. Statistics performed on transformed data are in orange and untransformed is in light blue. The table is organized by figure number, in chronological order. Statistical tests are listed and transformation equation provided, where applicable. When a paired t-test is performed on paired data, the results of the unpaired t-test are also given. When both paired and unpaired are potentially appropriate (in the case of paired data with ineffective pairing), both test results are given. Statistical test results in *italics* denote tests not presented within the figure.
DOI: https://doi.org/10.7554/eLife.39123.024

• Transparent reporting form
DOI: https://doi.org/10.7554/eLife.39123.025

## Data availability

All data generated or analyzed during this study are included in the manuscript and supporting files.

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
