## [Decision Letter]

[Editors’ note: this article was originally rejected after discussions between the reviewers, but the authors were invited to resubmit after an appeal against the decision.]

Thank you for submitting your work entitled "Potentiation of P2RX7 as a Host-Directed Strategy for Control of Mycobacterial Infection" for consideration by *eLife*. Your article has been reviewed by four peer reviewers, one of whom is a member of our Board of Reviewing Editors, and the evaluation has been overseen by a Reviewing Editor and a Senior Editor. The following individuals involved in review of your submission have agreed to reveal their identity: Amy Barczak (Reviewer #4).

Our decision has been reached after consultation between the reviewers. Based on these discussions and the individual reviews below, we regret to inform you that your work will not be considered further for publication in *eLife*.

Reviewers generally expressed enthusiasm for the elegant study design and clear and logical presentation. They noted the novel and interesting in vivo imaging methods, applied in a creative way to assess host directed effects. There was also enthusiasm for attempts made to determine the mode of action for clemastine. However, there were several notable shortfalls that preclude publication in *eLife*. These include the relatively marginal effect of clemastine, lack of clarity regards statistical evaluation and significance, no attempts to correlate florescent readouts with CFUs, and notable weaknesses in the inflammasome activation component of the study.

*Reviewer #1:*

Summary:

In their submission, the authors describe a study that uses zebra fish, at the transparent larval stage, to search for new host directed therapies (HDTs) for tuberculosis. They start by screening 1200 FDA approved drugs in a survival assay, with Mycobacterium marinum-infected in zebra fish, with a reduction in bacterial load as the measured outcome, and identify 23 promising candidates for further study. They focus on a subset of compounds that have host-directed activity rather than direct bactericidal activity per se and settle on further characterising clemastine, as it is inexpensive and widely available. The authors demonstrate that during infection with *M. marinum* in zebra fish, clemastine was able to reduce bacterial burdens in infect larvae in a dose-dependent manner. Given that this compound had no demonstrable antibacterial activity, this clearance was attributed the host-directed effects of clemastine. To further probe what aspect of host metabolism this drug disturbed, the authors treated infected larvae that lack macrophages, using an *irf8* mutant that is unable to generate macrophages. In this case, bacterial burdens were higher and clemastine had no effect, suggesting that clemastine targeted macrophage function. To dissect these effects in vivo, the authors conducted imaging of infected zebrafish where macrophages and bacteria were labelled with differentiating colours and demonstrate that clemastine reduced the intra-macrophage bacterial load. Next, the authors probe whether clemastine affects phagosome acidification and report no significant difference in acidification upon clemastine treatment. To further study the effects of clemastine on host cells, the authors assess its effect on the P2RX7 receptor. They generate mutant larvae defective for domains of this receptor and combine this with a calcium reporter, followed by the use of light sheet microscopy to study real-time calcium dynamics in vivo. Using this approach, the authors demonstrate that clemastine treatment results in an increase of calcium flashes, an effect that was abrogated in P2RX7 mutants. Consistent with this, clemastine had negligible activity in P2RX7 mutants, an observation further augmented with experiment using a P2RX7 inhibitor. Following these observations, the authors then assess the effect of clemastine on inflammasome activation by using an RD1 mutant of *M. marinum*, that lacks the specialized secretion apparatus required for mycobacterial interaction with host inflammation pathways. Clemastine had no effect in this case. To further probe inflammasome activation, the authors use a mutant defective in Asc, the adaptor required for inflammasome activation, and clemastine was ineffective in this background. Finally, the authors demonstrate that clemastine is effective in mature granuloma explants from adult zebrafish. This is a well-executed, clearly presented study. Some points detract; these are given below.

Essential revisions:

1) Videos require some detailed annotation and processing to outline the larvae and point out the bacteria, sometimes it's hard to figure out what is happening. More detailed legends would be useful. The stills in the Supplementary information do help, but without these the videos are not really useful and perhaps should be replaced by a series of more closely spliced still images. This is especially applicable with the calcium flashes, where it is harder to compare the rates of flashing to the control. Some thought should be given to this. Perhaps creating a composite movie of the control and clemastine treated fish side-by-side would be useful?

2) Confirmation of the various mutants generated in zebrafish seems to be missing; either Southern blot or Western blots indicating that protein domains were removed. Some form of confirmation should be provided.

3) A model that pulls all the observations together will enhance the readability of the manuscript.

*Reviewer #2:*

In this very elegant study, Matty et al. use the *M. marinum* / zebrafish model as an in vivo screening tool to identify host-directed FDA approved drugs that affect intracellular mycobacterial growth. After counter-screening pathogen-targeting molecules, they select and confirm Clemastine as the most potent host-directed candidate, and go on to decipher its mechanism of action, showing that macrophages are required, and that clemastine acts as an agonist of the P2RX7 receptor leading to increased intracellular calcium flashes which in turn activate the inflammasome known to control bacterial growth. The manuscript is very clearly written, the research strategy follows a logical flow and the experiments are thoroughly executed; the concern mostly lies in the size of the clemastine effect. Review by a statistics expert is recommended.

Essential revisions:

In the initial screen, it is somewhat surprising that several drug classes known to be active against *M. marinum* (https://doi.org/10.1186/s12941-016-0145-1) didn't emerge as primary hits: rifamycins, aminoglycosides, macrolides, trimethoprim. Could the authors comment on this?

Why use INH at a very high concentration (200 ug/mL) as a positive control since it as little to no activity against non-TB mycobacteria given that it's a prodrug requiring intrabacterial activation? There are many more potent alternatives, see above.

The size of the clemastine-mediated effect is rather small (two-fold effect at most across assays, conditions and experiments), triggering a few questions as to the read-outs and statistical analyses.

- How do fluorescence units relate to bacterial burden in zebrafish? Has this been calibrated? Could the authors explain this in subsection *“*Live imaging and quantification of bacterial burden”, beyond "Bacterial burden is presented as the product of area and mean intensity of bacterial fluorescence."?

- Were the non-transformed or log-transformed fluorescence values used in the statistical analyses? If the latter, could the author also include the p-values of the non-transformed data?

- In some experiments, the differences between what is declared as statistically significant and what isn't are subtle. This is best illustrated by Figure 2B and 2F where the visual impression doesn't clearly convey the message. While the burden is higher in macrophage-free zebrafish, the clemastine effect or lack thereof can't be assessed from these images.

In the Discussion section, could the authors mention and comment on the recent paper by Mawatwal et al., (2017): "Calcimycin mediates mycobacterial killing by inducing intracellular calcium-regulated autophagy in a P2RX7 dependent manner". What are the implications of their findings in the context of this study?

Reviewer #3:

In this manuscript the authors describe an in vivo whole animal screen of 1200 FDA-approved drugs to identify novel host-directed therapeutics with activity against mycobacteria. They identify clemastine as a drug that restricts replication of the zebrafish pathogen Mycobacterium marinum in vivo (but not in broth). Although clemastine is better known as an anti-histamine, the efficacy of the drug in vivo is proposed to be due to 'off-target' agonism of the fish P2RX7 ion channel that results in a protective response involving activation of a host immune 'inflammasome' pathway.

There are several nice aspects to the work. The manuscript is well written, and the data are logically and clearly presented. To my knowledge, this is the first in vivo screen for anti-mycobacterial drugs. This is significant because compounds with in vitro efficacy often fail in vivo. In addition, the authors also take advantage of the optical transparency of zebrafish embryos to perform some nice imaging analyses of infected animals, particularly the calcium flux experiments in Figure 3.

Despite these positive aspects, I have two major reservations about the potential impact of the work: (1) the effects observed are small and some of the statistical interpretations are problematic (see below); and (2) although the authors propose their observations are relevant to human tuberculosis, I do not find this sufficiently convincing in the absence of experiments with the actual human pathogen (*M. tuberculosis*) in mammals with lungs and adaptive immunity (which is absent in the embryos used here). In addition, mammalian/*Mtb* models often use survival or at least animal health as an endpoint 'gold standard', whereas only bacterial burdens are shown in the present manuscript. Because I suspect there will be some sensitivity with respect to this point, I should be clear that I am not saying the zebrafish is a poor model for *M. tuberculosis* lung infections; it may or may not be, and whether it is or not likely differs in different scenarios. I am merely saying that the relevance has not been established in this specific case, and this is a problem because such relevance is one of the authors' major claims for the significance of their work.

Essential revisions:

1) The magnitude of the effects of clemastine are very modest and appears to vary from experiment to experiment. The effect in Figure 2 is said to be approximately ~60% reduction (i.e., ~2-fold), but in other experiments (e.g., Figure 5A,B, Figure 6B, etc.), the effect appears to be even less. Obviously, there is no objective threshold that determines when the magnitude of a biological effect is sufficiently large for a journal such as *eLife*. It is a judgment call; and this reviewer's judgment is the magnitude of the effects will affect the impact of this work.

2) I am having trouble with the way statistics are used to support some of the main claims of the paper. My trouble relates to the difference between showing an observed difference is not significant versus showing that there is no difference. For example, in Figure 4, the authors seek to establish one of the main claims of the paper, namely that the effect of clemastine on bacterial burden is p2rx7 dependent. In fact, I believe what is shown is that there is no evidence that the drug has an effect in p2rx7 knockouts, which is quite a different thing (and could in fact be claimed even if no experiment had been done at all). Indeed, it appears to me that there may be an effect of the drug in p2rx7-deficient animals, even if the difference didn't reach a similar degree of statistical significance as in WT (perhaps due to the lower n?). Instead of reporting that there is no statistically significant effect of the drug in p2rx7 knockouts, I think it would be preferable to arrange the statistical testing in such a way that it is possible to conclude that the effect of the drug (fold restriction of bacterial burdens?) is statistically different in WT vs p2rx7 knockouts. In addition, is the Clemastine+P2RX7+ group significantly different than the Clemastine+P2RX7- group?

*Reviewer #4:*

In this work, Matty et al. use a zebrafish model of *M. marinum* infection to screen a bioactive library of small molecules to identify novel host-directed anti-mycobacterial agents. This work is novel in that it represents the first whole organism screening approach to identifying novel anti-TB therapies, and successfully identifies a drug in clinical use for other indications that has not previously been identified as restricting mycobacterial growth. Although antimycobacterial activity is not likely through its primary annotated target, the authors then identify a probable target and perform genetic studies to validate the target. The interpretation of the RD1 experiment results and presumed link between RD1 and the inflammasome is less convincing, with concerns as detailed below. The inclusion of the ex vivo granuloma model to evaluate longer term drug efficacy is both interesting and novel. Overall the study is well executed and in my opinion merits publication.

Essential revisions:

1) RD1 data and link to the inflammasome: The authors show that clemastine does not further restrict the growth of an RD1 mutant, which lacks ESX-1 and thus fails to permeabilize the phagosome. This is interpreted as demonstrating that the drug only has efficacy if the bacterium has access to cytosolic contents, which is then used to justify thinking about how the inflammasome might be involved. I think this is problematic on two fronts: (1) An attenuated mutant (the RD1 mutant) might not be further growth-restricted by a host-directed drug for many reasons other than cytosolic access. I think (at minimum) testing the effect of clemastine against an attenuated mutant still able to access the cytosol would be necessary to begin to make that interpretation (2) The logical bridge between the requirement for cytosolic access and inflammasome activation is a weak one. The literature is divided on whether ESX-1 (i.e. cytosolic access of the bacterium) is required for activation of the two inflammasomes shown to be relevant in TB- AIM2 and NLRP3. While older literature suggested that activation of NLRP3 required ESX-1, the paper (PMID: 26048138) that best dissected out the relative contributions of NLRP3 and AIM2 to the overall inflammasome response to TB found that while ESX-1 was required for AIM2 activation, it was not required for NLRP3 activation. Further, permeabilizing the phagosome is known to be required for triggering other pathways, among other effects- to me, even though the ASC mutant data shows clemastine has no effect, there are logical gaps in getting to that experiment. I think the paper would be logically stronger if the RD1 and inflammasome data were removed.

2) Lack of testing in combination with an established anti-mycobacterial drug: Host acting agents have relatively weak activity against *Mtb* when compared with standard antibiotics and will likely have to be used as adjuncts to other therapeutics. In some cases, drugs with activity against *Mtb* in host cells when used alone have been shown to compromise the activity of standard antibacterial agents when used in combination. An argument can be made that the standard for testing potential novel host-directed agents should include testing together with more standard antibiotics. If this type of model were not already established in the lab, these experiments would potentially go beyond 60 days - in which case I would still support publication of the manuscript without those experiments but would suggest that the potential for different effects in combination/need to test in combinations be addressed in the Discussion section.

[Editors’ note: what now follows is the decision letter after the authors submitted for further consideration.]

Thank you for choosing to send your work entitled "Potentiation of P2RX7 as a Host-Directed Strategy for Control of Mycobacterial Infection" for consideration at *eLife*. Your letter of appeal has been considered by a Senior Editor and a Reviewing editor, and we are prepared to consider a revised submission with no guarantees of acceptance.

In preparing your revision, we would like to draw your attention to some important points for consideration.

1) In the absence of rigorous statistical analysis, or the lack of a clear explanation as to how statistics were conducted, small (or even large) changes in a biological system are difficult to contextualize. We encourage you to consider this carefully in your data analysis. Regards statistics, pay particular attention to the correct choice of test to use, usually determined by the nature of the data, parametric versus non-parametric etc. Report these consistently in all figures, provide p-values for all comparisons (include the choice of test each time) and if necessary, include an expanded data analysis section in the Materials and methods section.

2) Consider more carefully how you would incorporate inflammasome activation as the mechanism into your revision. Reviewers raised several concerns regards the RD1 mutant. However, simply removing it from the study may weaken the mechanism. We encourage you to think further as to how you can address this concern.

3) We would like to affirm your sentiment that *eLife* values model systems as central to the scholastic pursuit of holistically understanding biological systems. The zebrafish model is no different in this case. However, it falls upon the authors to ensure that data presented in any manuscript are interpreted within the context of said model and not over-extended to humans or any other system without substantiating experiments. Please consider this carefully when preparing your revision and limit the interpretation to the model system that you are using, unless you have supporting/validation data from other systems.

---

## [Author Response]

[Editors’ note: the author responses to the first round of peer review follow.]

Reviewer #1:Summary:In their submission, the authors describe a study that uses zebra fish, at the transparent larval stage, to search for new host directed therapies (HDTs) for tuberculosis. […] This is a well-executed, clearly presented study. Some points detract; these are given below.Essential revisions:1) Videos require some detailed annotation and processing to outline the larvae and point out the bacteria, sometimes it's hard to figure out what is happening. More detailed legends would be useful. The stills in the supplementary information do help, but without these the videos are not really useful and perhaps should be replaced by a series of more closely spliced still images. This is especially applicable with the calcium flashes, where it is harder to compare the rates of flashing to the control. Some thought should be given to this. Perhaps creating a composite movie of the control and clemastine treated fish side-by-side would be useful?

We appreciate the reviewers’ concern with the videos in their previous form. To address this concern, we have provided each control and clemastine treatment video set as a split screen movie. We hope that the additional clarification in Video 1 (previously Video 1 and Video 2) shows more clearly that clemastine-treated macrophages display microbicidal activity, and we have added tracking of individual bacteria, marked at the end of the movie with solid or outlined arrowheads, to represent still-present or disappearing bacteria within individual macrophages.

For the calcium transients and this analysis, which appeared most difficult to discern, we have revamped the presentation in the videos. Video 3 and Video 4 (now Video 2) have been recreated as a split screen video to show that the wildtype and p2rx7 mutant calcium signaling is similar in uninfected animals at baseline, with frame-by-frame annotations for each flash counted. For the calcium video of infected animals (Video 3 and Video 4, previously Videos 5-8), we have also circled or boxed the regions of interest (ROIs) for the infected cells, which were counted for the number of calcium flashes in that cell during the course of the video. The boxes appear when the cell flashes and they are color coded: if a cell flashes only once, its box will be yellow and if a cell flashes more than once, the box will be the same color throughout the video. More detailed legends are also provided with these newly combined and annotated videos.

2) Confirmation of the various mutants generated in zebrafish seems to be missing; either Southern blot or Western blots indicating that protein domains were removed. Some form of confirmation should be provided.

We performed unambiguous Sanger sequencing as a confirmation of each deletion allele (Figure 3B and Figure 5—figure supplement 2A). Antibodies are not readily available for zebrafish P2rx7 and we were unable to find an antibody raised against mammalian P2RX7 that would cross react with zebrafish P2rx7. We regret that we cannot address this issue fully given the technology available to us but hope that the two distinct alleles each with the same phenotype suffice.

3) A model that pulls all the observations together will enhance the readability of the manuscript.

We agreed that a model would enhance the readability of the manuscript. A model has been provided as Figure 7, with a succinct summary of the findings of our work.

Reviewer #2:[…] Essential revisions:In the initial screen, it is somewhat surprising that several drug classes known to be active against M. marinum (https://doi.org/10.1186/s12941-016-0145-1) didn't emerge as primary hits: rifamycins, aminoglycosides, macrolides, trimethoprim. Could the authors comment on this?

We have included a sentence to comment on this point within the manuscript (Introduction). Briefly, the concentration at which the screen was performed (5 µM) is much lower than the concentration at which these drugs have been shown to have effects in zebrafish, perhaps due to permeability issues (Adams et al., 2011). Indeed, we note as part of the new antibiotic plus clemastine experiments in Figure 6 that the in vivo MIC of a drug like moxifloxacin is substantially higher in the zebrafish model than its in vitro MIC.

Why use INH at a very high concentration (200 ug/mL) as a positive control since it as little to no activity against non-TB mycobacteria given that it's a prodrug requiring intrabacterial activation? There are many more potent alternatives, see above.

This is an excellent point, and the concentration required for INH to be effective against *M. marinum* is unusually high and so it was a poor initial choice of control for lack of growth. Therefore, we have included an additional bacterial growth curve to address this, in which we use moxifloxacin at a concentration of 10 µg/mL as a positive control (Figure 6—figure supplement 3B). We also use moxifloxacin in additional new experiments with the granuloma model.

The size of the clemastine-mediated effect is rather small (two-fold effect at most across assays, conditions and experiments), triggering a few questions as to the read-outs and statistical analyses.

We point out above that within our model – a slow growing mycobacterium over the five-day infection possible in the zebrafish larva – this represents a fairly large and robust effect size. It is on a par with effect sizes described for host perturbations in a number of other zebrafish-mycobacterium papers over the past 15 years. However, we appreciate this point and, in the course of performing additional experiments, found even larger effects in the adult granuloma explant experiments, as documented in the revised Figure 6.

We examined burden in the adult granuloma explant model using a variety of assays: fluorescence readout; bioluminescent readout using a *M. marinum lux* strain that we constructed, and CFU plating.

Overall, in larvae and granuloma explants, the effect size of clemastine treatment was generally between 0.5 log_10_ and 1.4 log_10_, or between 3-fold and 27-fold in most experiments. We have also added 16S rRNA-based analysis to complement the fluorescence quantification. In larvae and adult granulomas, we found good concordance with fluorescence values. And CFU assays in the adult granuloma model showed consistent and robust effects of clemastine on mycobacterial burden.

Finally, to show that clemastine can be combined productively with antibiotics, we have included experiments in ex vivo granulomas in which co-treatment with clemastine makes an antibiotic more effective (Figure 6F). We have also tried to clarify statistical analyses by providing more detailed descriptions of the analyses performed and the p-values presented therein.

*- How do fluorescence units relate to bacterial burden in zebrafish? Has this been calibrated? Could the authors explain this in subsection “*Live imaging and quantification of bacterial burden”*, beyond "Bacterial burden is presented as the product of area and mean intensity of bacterial fluorescence.”?*

Others have shown that fluorescence units of bacteria relate directly to CFU (Adams et al., 2011; Takaki et al., 2013; Walton et al., 2018). To further confirm these findings in our specific studies, we have also provided 16s rRNA qPCR analysis of key experiments, which directly corroborate the findings in fluorescence readouts in larvae (Figure 4—figure supplement 1C and Figure 4—figure supplement 2C and Results section). We have further confirmed that the fluorescence readouts in granuloma explant models represent bacterial burden by luminescence (Figure 6D, F and Figure 6—figure supplement 3A, C and Discussion section) and analysis of colony forming units (CFU) (Figure 6E and Materials and methods section). Because these experiments are technically challenging and terminal endpoints, qPCR and CFU are provided for confirmation of fluorescence and luminescence readouts. Bacterial burden by fluorescence is calculated using ImageJ. Images are analyzed for the mean fluorescence and area of fluorescence above a threshold within the zebrafish (as determined by brightfield imaging). This threshold is empirically determined for each experiment (to ensure that every infect animal has a non-zero value) and kept constant between treatments. The product of the mean fluorescence and area is computed and presented as “Mm Fluorescence”. This was not clearly described the initial submission and has been added to the Materials and methods section.

- Were the non-transformed or log-transformed fluorescence values used in the statistical analyses? If the latter, could the author also include the p-values of the non-transformed data.

Due to the spread of the data, most bacterial data are presented as log_10_(Mm fluorescence) and statistics are performed on the data presented within each figure, i.e. if a figure shows transformed data, then the p-values shown are from the statistical tests on that data. We have provided statistics performed on both transformed and untransformed values in Supplementary file 2 and Supplementary file 3. In all experiments, the only circumstances under which transforming data changes the p-values are (1) when the variation among standard deviations changes, requiring a different statistical test correction, and (2) when there are values of ‘0’ among the results, as log_10_(0) is undefined. However, after proper thresholding, there were no ‘0’ ‘Mm fluorescence’ values as all animals had some degree of infection (see above).

- In some experiments, the differences between what is declared as statistically significant and what isn't are subtle. This is best illustrated by Figure 2B and 2F where the visual impression doesn't clearly convey the message. While the burden is higher in macrophage-free zebrafish, the clemastine effect or lack thereof can't be assessed from these images.

We agree that we should have done a better job presenting these images. The differences can appear subtle when showing the entire animal as an overlay of fluorescence and brightfield in a single two-dimensional image. We have now added arrowheads to mark the points of infection in each animal, so that the differences between clemastine-treated and control animals should be apparent. We have brightened the images in Figure 2F to contrast the bacteria and brightfield images. The brightening was performed equally as noted in the legend across the clemastine and DMSO-treated wildtype animals, is only for the purpose of better seeing the bacteria above the brightfield image and was not used for quantification. We think that bacterial burden and differences in the clemastine treated groups are now much more easily assessed. As noted within the figure legends, the animals that are used as example images are highlighted in the graph and are at or near the mean for the group.

In the Discussion section, could the authors mention and comment on the recent paper by Mawatwal et al., (2017): "Calcimycin mediates mycobacterial killing by inducing intracellular calcium-regulated autophagy in a P2RX7 dependent manner". What are the implications of their findings in the context of this study?

Thank you for pointing out the possibility for autophagy to be a mechanism through which clemastine reduces bacterial burden in a calcium-dependent manner. We have added this reference to the Discussion section and Results section as an introduction to a new set of experiments. We have explored the role for clemastine in autophagy-induced killing of mycobacteria in multiple ways and have found that clemastine has no effect on autophagic flux, using a zebrafish line which labels LC-3 puncta with GFP (He et al., 2009). In this line, we observed the same number of GFP: LC-3 puncta decorating intracellular bacteria in DMSO and clemastine-treated animals (Figure 5—figure supplement 2D). We have also observed that clemastine is still effective in zebrafish lacking critical autophagy-related genes, including ATG5, ATG7, ATG12, and Beclin1 (data not shown because these zebrafish mutants were provided to us by collaborators to test but are not yet published). In this case, we think that clemastine’s mechanism of action is independent of autophagy.

Reviewer #3:[…] Despite these positive aspects, I have two major reservations about the potential impact of the work: (1) the effects observed are small and some of the statistical interpretations are problematic (see below); and (2) although the authors propose their observations are relevant to human tuberculosis, I do not find this sufficiently convincing in the absence of experiments with the actual human pathogen (M. tuberculosis) in mammals with lungs and adaptive immunity (which is absent in the embryos used here). In addition, mammalian/Mtb models often use survival or at least animal health as an endpoint 'gold standard', whereas only bacterial burdens are shown in the present manuscript. Because I suspect there will be some sensitivity with respect to this point, I should be clear that I am not saying the zebrafish is a poor model for M. tuberculosis lung infections; it may or may not be, and whether it is or not likely differs in different scenarios. I am merely saying that the relevance has not been established in this specific case, and this is a problem because such relevance is one of the authors' major claims for the significance of their work.

We appreciate the reviewer’s critique and enthusiasm for the novelty of the in vivo screen for anti-mycobacterial drugs that may be host directed. We have addressed the reviewer’s two major reservations throughout.

1) Effect size and statistical interpretations.

We point out above that within our model – a slow growing mycobacterium over the five-day infection possible in the zebrafish larva – this represents a fairly large and robust effect size for a host-directed monotherapy. It is on a par with effect sizes described for host perturbations in a number of other zebrafish-mycobacterium papers over the past 15 years, as well as host perturbations in other animal models. However, we appreciate this point. We note that, in the course of performing additional experiments for this revision, we confirmed the burden differences in a number of independent ways: 16S rRNA analysis; construction of a *lux* bioluminescent strain to measure burden in granuloma explants; and CFU measurement in treated granulomas. We noted even larger effects in the adult granuloma explant experiments as now measured by CFU and luminescence (median effect of 1.4 log_10_ reduction in burden over seven days relative to vehicle) as documented in the revised Figure 6.

We have also now included more robust statistical descriptions for the analyses we performed within the figure legends throughout, including, in supplementary data, adding parallel analysis of both log transformed and non-transformed data and providing two supplementary tables (Supplementary file 2 and Supplementary file 3) comprehensively documenting all p values throughout. We have also performed specific analyses requested by the reviewer, described below.

We have also added new experiments showing that clemastine can be used to enhance the effect of an antibiotic in the ex vivo granuloma model (Figure 6F and Figure 6—figure supplement 3C).

2) Concerns about the zebrafish

We recognize and appreciate that our findings are within the context of an animal model of mycobacterial infection and have emphasized the new biological insights obtained using a natural host-pathogen pairing and a whole animal in vivo drug screen. We have restructured the Introduction and Discussion section to focus on those findings within this context, and while we do believe that animal models are useful in understanding conserved fundamental features of mycobacterial infection, have been careful to emphasize that our current findings are within this system.

Essential revisions:1) The magnitude of the effects of clemastine are very modest and appears to vary from experiment to experiment. The effect in Figure 2 is said to be approximately ~60% reduction (i.e., ~2-fold), but in other experiments (e.g., Figure 5a, 5b, 6b, etc.), the effect appears to be even less. Obviously, there is no objective threshold that determines when the magnitude of a biological effect is sufficiently large for a journal such as eLife. It is a judgment call; and this reviewer's judgment is the magnitude of the effects will affect the impact of this work.

The effect is consistent and usually a 3-10 fold reduction (0.5 log_10_ to 1 log_10_) over a relatively short time period, as discussed above. To further support our findings, we have added additional methods to detect changes in bacterial burden. Specifically, we have used 16S rRNA qPCR analyses to confirm that clemastine reduces bacterial burden by an independent measure in larvae but fails to do so in *p2rx7* mutant animals (Figure 4—figure supplement 1C) and RD1 mutant bacteria (Figure 5—figure supplement 1C). Further, in new data from the granuloma explant model, clemastine’s effect appears greater (median reduction of 1.4 log_10_ when measuring by CFU plating). In addition, we show similar results over seven days of treatment against infections using the new *lux* strain.

2) I am having trouble with the way statistics are used to support some of the main claims of the paper. My trouble relates to the difference between showing an observed difference is not significant versus showing that there is no difference. For example, in figure 4, the authors seek to establish one of the main claims of the paper, namely that the effect of clemastine on bacterial burden is p2rx7 dependent. In fact, I believe what is shown is that there is no evidence that the drug has an effect in p2rx7 knockouts, which is quite a different thing (and could in fact be claimed even if no experiment had been done at all). Indeed, it appears to me that there may be an effect of the drug in p2rx7-deficient animals, even if the difference didn't reach a similar degree of statistical significance as in WT (perhaps due to the lower n?). Instead of reporting that there is no statistically significant effect of the drug in p2rx7 knockouts, I think it would be preferable to arrange the statistical testing in such a way that it is possible to conclude that the effect of the drug (fold restriction of bacterial burdens?) is statistically different in WT vs p2rx7 knockouts. In addition, is the Clemastine+P2RX7+ group significantly different than the Clemastine+P2RX7- group?

This point is well-taken. We have taken the reviewer’s suggestion about this analysis in Figure 4—figure supplement 1B and 1C and Figure 6—figure supplement 1B. We felt it was important to show actual unprocessed data first (without the fold-change processing), so have left the analysis and comparisons in Figure 4A intact and still do note that there is no statistically significant effect of the drug in the *p2rx7* knockout. However, as the reviewer suggested, in Figure 4—figure supplement 1B, we now show that the fold-reduction in burden with clemastine is significantly lower in WT vs. *pr2rx7* knockouts (p=0.001). We also performed these same analyses on granuloma explants and observe similar effects (Figure 6—figure supplement 1B). In addition, in response to the reviewer’s point 3 below, we performed 16S rRNA analysis on pooled larvae (four groups: WT vehicle, WT treated, p2rx7 vehicle, p2rx7 control) as a correlate of burden independent of fluorescence (Figure 4—figure supplement 1C) and found reduced burden in the WT animals and no effect in the p2rx7 knockouts.

Additionally, we have performed similar analyses for data supporting our findings in RD1 mutant bacteria and *asc* mutant animals. These new figures can be found in Figure 5—figure supplement 1C and Figure 5—figure supplement 2C. We show that clemastine reduces the fold change in bacterial burden compared to vehicle controls in wildtype infections more than RD1 mutant infections (Figure 5—figure supplement 1C). These data are also corroborated with 16S rRNA qPCR analysis within the same figure. Finally, we use this reviewer’s suggestion for additional analysis based on fold change to confirm that clemastine reduces the fold change in bacterial burden in wildtype animals compared to *asc* mutants (Figure 5—figure supplement 2C). Thus, not only is there a lack of a significant clemastine effect on infections with RD1 bacterial mutants, *asc* mutants, or *p2rx7* mutants, as shown previously, but the additional analysis suggested by the reviewer also reveals a statistically significant difference in the fold restriction of bacterial burden by clemastine between WT animals and each of these three genotypes. We hope these additional analyses, new experiments, and independent approaches to quantitate burden address this major concern.

Reviewer #4:[…] Essential revisions:1) RD1 data and link to the inflammasome: The authors show that clemastine does not further restrict the growth of an RD1 mutant, which lacks ESX-1 and thus fails to permeabilize the phagosome. This is interpreted as demonstrating that the drug only has efficacy if the bacterium has access to cytosolic contents, which is then used to justify thinking about how the inflammasome might be involved. I think this is problematic on two fronts: (1) An attenuated mutant (the RD1 mutant) might not be further growth-restricted by a host-directed drug for many reasons other than cytosolic access. I think (at minimum) testing the effect of clemastine against an attenuated mutant still able to access the cytosol would be necessary to begin to make that interpretation (2) The logical bridge between the requirement for cytosolic access and inflammasome activation is a weak one. The literature is divided on whether ESX-1 (i.e. cytosolic access of the bacterium) is required for activation of the two inflammasomes shown to be relevant in TB- AIM2 and NLRP3. While older literature suggested that activation of NLRP3 required ESX-1, the paper (PMID: 26048138) that best dissected out the relative contributions of NLRP3 and AIM2 to the overall inflammasome response to TB found that while ESX-1 was required for AIM2 activation, it was not required for NLRP3 activation. Further, permeabilizing the phagosome is known to be required for triggering other pathways, among other effects- to me, even though the ASC mutant data shows clemastine has no effect, there are logical gaps in getting to that experiment. I think the paper would be logically stronger if the RD1 and inflammasome data were removed.

While the RD1 data and ASC data are robust, we agree that this is a complex part of the field that may be difficult to fully resolve within the time-frame and scope of a revision. To strengthen this aspect of the manuscript, we have performed new experiments that (1) address the specificity of the RD1 effect and (2) provide some evidence (beyond genetic dependence on *asc*) of clemastine-treated granulomas exhibiting increased inflammasome-associated activity.

To address the reviewer’s first concern, we tested the effect of clemastine against other mycobacterial mutants. Many of the attenuated mutants we tried (e.g. *erp*) were so attenuated that they were not useful in these assays, as they were being cleared on their own. However, we found that a bacterial mutant with a transposon insertion in *cmaA2* was attenuated in our model. CmaA2 encodes a mycolic acid trans-cyclopropane synthetase (Glickman et al., 2001). While we did not seek to define the mechanism for *cmaA2* mutant attenuation in our model, we show that these bacteria fail to expand normally in vivo. Despite their attenuation and low bacterial burden, clemastine treatment further reduces bacterial burden by ~0.5 log_10_ in *cmaA2* mutants (Figure 5—figure supplement 1B-C), in contrast to the RD1 results. In addition, we used a reciprocal approach in which we infected with a high dose of RD1 mutant bacteria to show that even with a higher burden of infection, clemastine remains ineffective against RD1 mutant bacteria (Figure 5—figure supplement 1B-C), described in subsection “Clemastine is effective in established infections”.

We agree that the literature is complex and sometimes contradictory on how, whether, and when ESX-1 and/or cytosolic access is required for activation of inflammasomes during mycobacterial infection. The finding that clemastine was ineffective against RD1 mutant bacteria led us to investigate the inflammasome as a mechanism through which clemastine can reduce bacterial burden, as we know that ESX-1 secretion systems have been implicated in both the activation and inhibition of inflammasomes during mycobacterial infection. The manuscript does not seek to resolve the role of RD1/ESX1 in inflammasome activation. Rather, it was the impetus to examine this pathway and the *asc* mutants and find that clemastine was ineffective in both *p2rx7* and *asc* mutants. We also have added Figure 6C and Figure 6—figure supplement 2A-C, in which we devised a protocol for quantitating FLICA-reagent fluorescence in granuloma explants after clemastine treatment, suggesting increased inflammasome activity after clemastine treatment. Unfortunately, the administration and/or sensitivity of this reagent did not allow us to perform these experiments in larvae.

To bolster the case for specificity, we also added negative data investigating a potential role for autophagy in mediating clemastine’s effects. In new Figure 5—figure supplement 2D, we see no difference in LC3 positive puncta upon clemastine treatment using a previously published zebrafish line (He et al., 2009).

2) Lack of testing in combination with an established anti-mycobacterial drug: Host acting agents have relatively weak activity against Mtb when compared with standard antibiotics and will likely have to be used as adjuncts to other therapeutics. In some cases, drugs with activity against Mtb in host cells when used alone have been shown to compromise the activity of standard antibacterial agents when used in combination. An argument can be made that the standard for testing potential novel host-directed agents should include testing together with more standard antibiotics. If this type of model is not already established in the lab, these experiments would potentially go beyond 60 days- in which case I would still support publication of the manuscript without those experiments but would suggest that the potential for different effects in combination/need to test in combinations be addressed in the Discussion section.

We wholeheartedly agree that host-acting agents typically have weak activity against *Mtb* when compared to standard of care antibiotics. HDTs would be used in conjunction with the standard of care in any human trial or treatment. While the time-frame of the revision precluded a comprehensive examination of front-line antibiotics, we examined the effects of clemastine in conjunction with the antibiotic moxifloxacin, selected in part because it displayed reduced efficacy in the granuloma model (10-fold higher MIC relative to in vitro MIC – Cronan et al., 2018). In combination with moxifloxacin, clemastine co-treatment is more effective than antibiotic alone (Figure 6F and Figure 6—figure supplement 3C and subsection “Transgenic lines”).

[Editors’ note: the author responses to the re-review follow.]

We would like the thank the reviewers for their thorough and thoughtful critique of our manuscript. We are pleased to have had the opportunity to present the information more clearly and perform new experiments to address reviewer concerns. We have added substantial new experimental data, including validating findings about bacterial burden with three additional independent approaches and find even more robust effects in the established granuloma model (now using three independent approaches to quantify burden) than we had presented previously in the larval infection model. We have addressed statistical concerns more comprehensively throughout, and strengthened the findings around RD1 and inflammasome activation using new mycobacterial mutants, additional experiments, and a flow-cytometry-based analysis of clemastine’s effect on established granulomas. In addition, as requested by the reviewers, we have shown efficacy of clemastine in combination with a traditional antibiotic. And we have reworded significant aspects of the discussion to make clear the novel biological findings and approach within the context of the zebrafish-mycobacterial infection system, a natural host-pathogen pairing. In summary, we hope we have addressed the reviewers’ concerns through a number of additional experiments, thoughtful rewording, and careful statistical analysis throughout.

As presented below in more detail, we have addressed each of these points with new experiments.

Regarding the effect size of clemastine initially reported (~0.5-1.0 log_10_ by fluorescence), within our model – a slow growing mycobacterium over the five-day infection possible in the zebrafish larva – this represents a fairly large and robust effect size. It is on a par with effect sizes described for host perturbations in a number of other zebrafish-mycobacterium papers over the past 15 years (e.g. Clay et al., 2007; Tobin et al., 2010; Tobin et al., 2012; Roca and Ramakrishnan, 2013; Oehlers et al., 2015 among others). Many of these findings have been validated in other mycobacterial infection models and in human populations. In addition, we performed new experiments to validate the robustness of the effect.

Importantly, the development of a new granuloma explant model has allowed us to examine treatment over longer time-frames with established infections as well as enabled new approaches to quantifying burden (thereby independently validating the fluorescence-based results) that are not possible in the larvae. In new experiments, we found that clemastine’s effect on burden measured by CFU was a 27-fold (1.4 log_10_) median reduction over seven days of treatment in established infections. We also developed a new lux-based strain for longitudinal quantitation in established granulomas and obtained similar results.

In the revised manuscript, we have further clarified all statistical tests. All significant p-values are presented within the figure and all non-significant p-values are described in each figure legend. Additionally, all p-values (for both transformed and untransformed data) are presented in Supplementary file 2 and Supplementary file 3. New experiments are included to correlate fluorescent readouts with CFU, 16S bacterial rRNA, and a new mycobacterial strain that expresses Lux to provide an even more accurate and independent readout of bacterial counts within nascent and established infections. Using the granuloma explant model and flow cytometry, we have further expanded our investigation of the role for clemastine in activation of the inflammasome; clemastine treatment efficacy is not only dependent on *asc* but, in new data, we detect enhanced activation of caspase-1 via a FLICA reagent and flow cytometry of established granulomas treated with clemastine.

1) In the absence of rigorous statistical analysis, or the lack of a clear explanation as to how statistics were conducted, small (or even large) changes in a biological system are difficult to contextualize. We encourage you to consider this carefully in your data analysis. Regards statistics, pay particular attention to the correct choice of test to use, usually determined by the nature of the data, parametric versus non-parametric etc. Report these consistently in all figures, provide p-values for all comparisons (include the choice of test each time) and if necessary, include an expanded data analysis section in the Materials and methods section.

We have more clearly explained how we conducted our statistical analyses and included much more information about the analyses we did perform. We have reported all p values for both transformed and untransformed data as requested by one of the reviewers and this is now included comprehensively in new Supplementary file 2 and Supplementary file 3.

2) Consider more carefully how you would incorporate inflammasome activation as the mechanism into your revision. Reviewers raised several concerns regards the RD1 mutant. However, simply removing it from the study may weaken the mechanism. We encourage you to think further as to how you can address this concern.

To further address these concerns, and as detailed below and above, we performed additional experiments that were supportive of this model. These included generating and assessing a new mycobacterial mutant to rule out the possibility of burden-dependent efficacy of clemastine; calibrating RD1 infection levels to address this same issue; and developing a new protocol in established granulomas to quantitate caspase-1 activation via a FLICA reagent upon clemastine treatment. More details are provided in the point-by-point response to reviewers.

3) We would like to affirm your sentiment that eLife values model systems as central to the scholastic pursuit of holistically understanding biological systems. The zebrafish model is no different in this case. However, it falls upon the authors to ensure that data presented in any manuscript are interpreted within the context of said model and not over-extended to humans or any other system without substantiating experiments. Please consider this carefully when preparing your revision and limit the interpretation to the model system that you are using, unless you have supporting/validation data from other systems.

We appreciate that our findings come within the context of an animal model of mycobacterial infection and have emphasized the new biological insights obtained using a natural host-pathogen pairing and an in vivo drug screen. We have restructured the Introduction and Discussion section to focus on those findings within this context, and while we do believe that animal models of mycobacterial infection are useful in understanding conserved fundamental features of mycobacterial infection, have been careful to emphasize that our findings are within this particular system.